# Rare flood scenarios for a rapidly growing high-mountain city: Pokhara, Nepal

Melanie Fischer[1], Jana Brettin[1], Sigrid Roessner[2], Ariane Walz[1], Monique Fort[3], Oliver Korup[1,4]

[1]Institute of Environmental Science and Geography, University of Potsdam, Potsdam, Germany

[2]Helmholtz Centre Potsdam, GFZ German Research Centre for Geosciences, Potsdam, Germany

[3]Département de Géographie, Université Paris Cité, Paris, France

[4]Institute of Geosciences, University of Potsdam, Potsdam, Germany

*Correspondence to*: Melanie Fischer melaniefischer@uni-potsdam.de)

**Abstract**

Pokhara (c. 850 m a.s.l.), Nepal's second largest city, lies at the foot of the Higher Himalayas and has more than tripled its population in the past three decades. Rapidly expanding built-up areas are high in demand for construction materials and several informal settlements cater to unregulated sand and gravel mining in the Pokhara valley's main river, the Seti Khola. This river is fed by the Sabche glacier below Annapurna III (7,555 m a.s.l.), some 35 km upstream of the city, and traverses

one of the steepest topographic gradients in the Himalayas. In May 2012 a sudden flood caused >70 fatalities and intense damage along this river and rekindled concerns about flood-risk management. We estimate the flow dynamics and inundation depths of flood scenarios using the hydrodynamic model HEC-RAS. We simulate the potential impacts of peak discharges from 1,000 to 10,000 $m^3 s^{-1}$ on land cover based on high-resolution Maxar satellite imagery and OpenStreetMap data (buildings and road network). We also trace the dynamics of two informal settlements near Kaseri and Yamdi with high potential flood

impact from RapidEye, PlanetScope, and Google Earth imagery of the past two decades. Our hydrodynamic simulations highlight several sites of potential hydraulic ponding that would largely affect these informal settlements and sites of sand and gravel mining. These built-up areas grew between three and twentyfold, thus likely raising local flood exposure well beyond changes in flood hazard. Besides these drastic local changes, about 1% of Pokhara's urban built-up area and essential rural road network is in the highest hazard zones highlighted by our flood simulations. Our results stress the need to adapt early-

warning strategies for locally differing hydrological and geomorphic conditions in this rapidly growing urban watershed.

**1 Introduction**

Many mountain communities and their infrastructure have become exposed and vulnerable to natural hazards (Fort, 2015; Hock et al., 2019). The Hindu-Kush Himalayas, home to some 200 million people and "water tower" to 1.3 billion people living downstream (Immerzeel et al., 2010; Schild, 2008), have seen rapid population growth, expanding road networks, shifts

from agriculture to tourism as the main economic revenue, and the rise of hydropower projects (Hock et al., 2019; Schwanghart et al., 2018; Sidle and Ziegler, 2012). Current projections of cryospheric change in this mountain belt foresee a continued total glacier-mass wastage of up to 64 ± 5% by the end of the 21st century under the RCP8.5 scenario (Kraaijenbrink et al., 2017) as well as permafrost degradation (Bolch et al., 2019). These changes will likely result in a destabilisation of mountain slopes and increase in meltwater volumes stored in lakes impounded behind potentially unstable natural dams (Hock et al., 2019).

Thus, the potential for hazards caused from these instabilities, including sudden floods, is likely to increase in the future. The Pokhara valley in Nepal, home to the nation's second largest city, is a prime example of a Himalayan valley experiencing rapid socio-economic development. Emerging job opportunities in the tourism sector triggered the steep rise of Pokhara's population since the 1970s, fuelled by unabated migration from rural to urban areas (Rimal et al., 2015, 2018). Urbanisation pressure has also forced informal settlements of marginalised communities on the lowermost river terraces and floodplains of the valley's

main river, the Seti Khola (Fort et al., 2018; Thapa et al., 2022).

Apart from annual monsoonal floods, this river has a history of rare, extreme floods. On May 5, 2012, a hyperconcentrated flow killed 72 persons and destroyed roads, bridges, and drinking water pipelines in the northern Pokhara valley (Gurung et al., 2015; Gurung et al. 2021). The exact sequence of events remains debated, but may have been initiated by rock-slope failures from the western flank of the Annapurna IV massif at 7,525m a.s.l., observed by chance by a pilot (Hanisch et al., 2013; Kargel et al., 2013). Similar to the 2021 Chamoli disaster (Shugar et al., 2021), a highly mobile ice-rock avalanche may have transformed into an hyperconcentrated flow that hit Kharapani village (1,100 m a.s.l.) some 23 km downstream just half an hour later, causing most of the damage and fatalities with an estimated peak discharge of 8,400 $m^3 s^{-1}$ (Hanisch et al., 2013; Oi et al., 2014; SANDRP, 2014). Thanks to the pilot, a radio warning was issued, most likely preventing a higher death toll further downstream (Kargel et al., 2013).

Even larger catastrophic floods may have occurred in the Seti Khola in Medieval times, depositing much of the youngest sediment fill of the Pokhara valley during or shortly after large earthquakes, likely in the wake of major landslides or outbursts of glacier- and landslide-dammed lakes in the Annapurna Massif (Fort, 1987; Schwanghart et al., 2016; Stolle et al., 2017). Based on relict natural dams, Schwanghart et al. (2016) estimated that up to 1 $km^3$ of water could have been stored in the steep walled and sediment-filled Sabche Cirque some 35 km north of Pokhara city; outburst floods from this cirque could have released water at peak rates of up to 600,000 $m^3 s^{-1}$. Several authors agree that the cirque might spawn large floods along the Seti Khola in the future (Fort, 2010; Grandin et al., 2012; Gurung et al., 2021; Kargel et al., 2013; Lovell et al., 2018). The upper Seti Khola gorge below the cirque is a bottleneck prone to blockage by landslides detaching from the cirque walls, and might impound large amounts of water (Kargel et al., 2013). Further downstream, landslides triggered by monsoonal storms (Talchabhadel et al., 2018) could also form temporary dams that might fail catastrophically like in the Melamchi outburst flood in June 2021 (Petley, 2021). Fort (1987) and Kargel et al. (2013) reported that the Sabche Cirque stores large amounts of unconsolidated material to nourish floods and debris flows. Although not present currently, meltwater lakes could form and grow rapidly in the Sabche Cirque and may release GLOFs in the near future (Zheng et al., 2021). The Sabche glacier could also contribute to generating outburst floods as its surges could form potentially short-lived lakes (Lovell et al., 2018). Despite this evidence of past non-meteorological floods along the Seti Khola, appraisals of flood hazard have so far largely focused on the 100-year meteorological flood as estimated from rainfall data (Basnet and Acharya, 2019; Gurung et al., 2021; Shrestha et al., 2021). The associated risk estimates rely on land use and land cover (LULC) mapping and projections from Landsat data (Gurung et al., 2021; Rimal et al., 2015, 2018), floodplain mapping (Gurung et al., 2021; Rimal et al., 2015, 2018), and hydrodynamic modelling for selected reaches (Basnet and Acharya, 2019; Gurung et al., 2021).

We aim to expand on these studies by providing a comprehensive, semi-quantitative estimation of potential non-meteorological flood impacts on the city of Pokhara and surrounding regions. In our approach we recognise possibly inundated areas for different flood scenarios, the types of land cover and infrastructure most likely affected and the role of rapid urban expansion. We intersect the results of hydrodynamic modelling with land cover data on buildings and the road network, and highlight the rapid recent growth of two informal settlements. Our appraisal forms a building block for a more formal risk assessment that is currently curtailed by a lack of data on non-meteorological flood probabilities as well as exposed values. Nonetheless, our

study is one of the few to combine flood scenarios with land-cover and land-use changes and might aid both urban planning and anticipatory risk management in the Pokhara valley (Nussbaumer et al., 2014).

## 2 Study area

Pokhara is the second largest city of Nepal, capital of the Kaski District and the Gandaki Province, and lies at the southern foot of the 8-km high, seismically active Annapurna massif of the Higher Himalaya (Fort, 2010; Grandin et al., 2012). The
80 Seti Khola is Pokhara's main river and traverses one of the steepest topographic gradients in the Himalayas, originating at 3,700 m a.s.l. in the Sabche Cirque of the Annapurna massif and entering the city 850 m a.s.l. some 35 km downstream (Fig. 1). The headwaters are mainly fed by Sabche glacier, the only glacier with observed surges in the central Himalayas (Lovell et al., 2018). The Seti Khola is also fed by the Mardi Khola, entering the valley in the north-west, and the Phusre Khola, to the south-east of Pokhara.

Pokhara is built on a large (>120 km²) intramontane alluvial fan of the Seti Khola. The youngest of three depositional units of this fan is the 60-100 m thick Pokhara Formation that was formed by (post-)seismic sediment pulses in the $12^{th}$ to $14^{th}$ century CE (Fort, 2010; Schwanghart et al., 2016). These mostly unconsolidated gravel beds cap the more indurated, but undated, Ghachok Formation (Fort, 1987; Hormann, 1974). Over a course of 70 km, the Seti Khola cuts through its own sediments, forming broad, unpaired, and 100-m high terraces (Fort, 2010; Stolle et al., 2017). These cut-and-fill terraces alternate with
several short (<1 km), narrow (<10 m) but up to 90-m deep gorges, in resistant calcareous conglomerates of the Ghachok Formation and the bedrock of the Lower Himalayan Sequence (Stolle et al., 2019). Some of these gorges are connected to karst features like potholes, tunnels, and caves (Fort, 2010).

Climate in the Seti Khola catchment is distinctly seasonal. The summer monsoon (May to October) brings >80% of the annual precipitation of about 4000 mm per year (Gabet et al., 2004). Climate also strongly varies with topographic relief (Rimal et
al., 2018): the central Pokhara valley has a humid sub-tropical to humid temperate climate with mean monthly temperatures of 13°C to 26°C (Ross and Gilbert, 1999), whereas temperate to alpine climate characterises the Annapurna massif to the north (Rimal et al., 2015, 2018).

With a population of about 518,000 in 2021 (Central Bureau of Statistics, 2022) the city and its surrounding valley have seen rapid socio-economical changes following the construction of major transportation infrastructure since the 1970s, driven by
100 better access to higher standards of living, and inbound migration attracted by the growing tourism sector and new employment opportunities (Fort et al., 2018).

Between 1990 and 2013, the urban area increased by 30 km², or 2.6% of the total watershed, whereas cultivated land decreased by 2.5% (Rimal et al., 2015). Population migration dynamics are also reflected by an increase of 45% in areas classified as urban in 2010 when compared to Pokhara's municipal area in 1977 (Rimal, 2012; Rimal et al., 2015). The urban population
more than tripled since the 1990s (Rimal et al., 2015; United Nations Department of Economic and Social Affairs, 2019).

Many informal settlements formed on the lowermost river terraces or the floodplain of the Seti Khola (Fig. 1), where squatters largely rely on gravel and sand mining as an income source (Fort et al., 2018; Gurung et al., 2021; Stolle, 2018).

## 3 Data and methods

### 3.1 Overall approach

We analyse potential impacts from physically plausible magnitudes of sudden floods along a 40-km long reach of the Seti Khola. Data to inform our range of scenarios come from two non-meteorological floods in the Pokhara valley, and have estimated peak discharges differing by orders of magnitude: while the May 2012 flood involved between 1,000 and 12,300 m³ s$^{-1}$ (Kargel et al., 2013; Oi et al., 2012; SANDRP, 2014), much larger Medieval floods may have involved 45,000 to 600,000 m³ s$^{-1}$ (Schwanghart et al., 2016), judging from geomorphic flood markers. Reported peak discharges from non-meteorological

floods elsewhere in the Himalayas are 1,600 m³ s$^{-1}$ for the Dig Tsho GLOF of 1985 (Vuichard and Zimmermann, 1987), and 8,000 to 14,000 m³ s$^{-1}$ for the 2021 Chamoli disaster (Pandey et al., 2022; Shugar et al., 2021), and thus support our range of historic flood scenarios plausibly. Our assessment is built on combining scenario-based inundation modelling with current land use and land cover (LULC) and data of buildings and the road network to identify areas of potential flood impact (Fig. 2). We also selected two informal settlements close to the river and estimate how the growth of their built-up area between

2008 and 2021 affected their exposure to floods. Stream-gauge data is unavailable for the Seti Khola, hence we validated our model with mapped damage and sediment traces caused by the May 2012 flood (Appendix, Fig. A1).

### 3.2 Scenarios for numerical flood routing

We simulated ten flood scenarios with the Hydrologic Engineering Center's River Analysis System, HEC-RAS (version 5.0.7; https://www.hec.usace.army.mil/software/hec-ras/). HEC-RAS uses a step method for simulating steady, i.e. constant but

gradually varied, open-channel flow in one dimension – based on the Saint-Venant equations (Brunner, 2020a; Westoby et al., 2014). The standard step method computes water-level profiles between two river cross-sections that describe the geometry of the channel and the adjacent overbank areas by iteratively solving the Energy equation (Eq. 1):

$$Z_2 + Y_2 + \frac{a_2 V_2^2}{2g} = Z_1 + Y_1 + \frac{a_1 V_1^2}{2g} + h_e \tag{1}$$

For cross-sections 1 and 2, $Z_1$ and $Z_2$ describe their main channel bed elevations, $Y_1$ and $Y_2$ the corresponding flow depths, $V_1$ and $V_2$ the mean flow velocities, and $a_1$ and $a_2$ the weighting coefficients; $g$ is the gravitational acceleration. The energy head loss $h_e$ specifies the effects of friction, as well as expansion and contraction caused by the channel geometry. At each cross-section, Manning's equation is used to compute discharge $Q$ (Eq. 2):

$$Q = \frac{1}{n} A R^{2/3} S_f^{1/2}, \qquad\qquad\qquad\qquad\qquad\qquad\qquad\qquad (2)$$

where $A$ is cross-section area, $R$ is the hydraulic radius, $S_f$ is the energy gradient, and $n$ is a hydraulic loss (or roughness) coefficient (Brunner, 2020a).

Geometric data for our HEC-RAS runs were mainly derived from the commercial ALOS 3D digital elevation model (AW3D DEM), which has a vertical and horizontal resolution of <5 m and was projected to UTM Zone 44N (Fig. 2). We also acquired field data with a TruPulse 360 laser range finder and a Garmin eTrex handheld GPS during two campaigns in October of 2016 and 2019. We used these field data to correct some of the 572 DEM-derived cross-sections of the Seti Khola and its major

tributaries, especially in the narrow gorges (Fig. 1). The average spacing between cross-sections was 90 m (Appendix, Fig. A2). We estimated Manning's $n$ from manual mapping of land-cover classes based on 1-m resolution Maxar satellite data (ESRI and Maxar Technologies, 2022) following guidelines by Brunner (2020b). We also estimated Manning's $n$ at 61 field locations following methods by Arcement Jr and Schneider (1984) and Chow (1959). We defined ten discharge scenarios with peak discharge $1,000 \le Q_p \le 10,000$ m³ s⁻¹ in the main channel. This range covers monsoonal floods (1,500 to 2,300 m³ s⁻¹;

Basnet and Acharya, 2019) and also larger but rarer non-meteorological floods like the one in May 2012 (8,400 m³ s⁻¹; Oi et al., 2014). The steepness of the Seti Khola's upper reach led us to specify mixed flow conditions with the critical flow depth as the upper boundary condition, while we set a normal depth of 0.0065 as the lower boundary condition. We assumed a constant base flow of 100 m³ s⁻¹ (i.e. 10% of our lowest discharge scenario) in three ungauged tributaries, i.e. the Mardi Khola, Kali Khola, and Phusre Khola.

**3.3 Land-use and land-cover data**

Using 1-m Maxar satellite imagery acquired in 2020 (ESRI and Maxar Technologies, 2022), we manually mapped ten land cover (LC) and two land use (LU) classes in our model domain with an area of 136 km² (Table 1, Fig. 1).

We derived object-scale data on Pokhara city's buildings, as well as roads and footpaths from the OpenStreetMap (OSM) collaborative project (https://www.openstreetmap.org/) via the Overpass Turbo tool (https://overpass-turbo.eu/) to capture the

160 situation in September 2021.

We mapped built-up areas in two informal settlements from medium- to high-resolution satellite imagery of April 2008 (Google Earth), November 2012 (Google Earth and RapidEye), and November 2021 (Google Earth and PlanetScope). In these images, we also identified sand and gravel mining activities based on the presence of artificial groynes and gravel heaps at the active channel margins.

## 3.4 Potential flood impact analysis

We used a geospatial overlay of our modelled flood inundation boundaries with the LULC data to assess on a semi-quantitative basis the likely impacts of ten peak discharge scenarios. We defined ten flood hazard classes by assigning areas and objects of the smallest flood scenario ($Q_p$ = 1,000 m³ s$^{-1}$) to the highest hazard class (HC) 10. Conversely, the lowest hazard class 1 is attributed to areas and objects that would be inundated by the largest floods only ($Q_p$ = 10,000 m³ s$^{-1}$). In contrast to the total area flooded in a given scenario, these hazard classes do not overlap and label instead the relative increases in inundation area with each peak discharge scenario. Thus, HC 1 defines areas and objects that are potentially impacted by a flood with $Q_p$ >= 10,000 m³ s$^{-1}$, whereas HC 10 defines areas and objects that would be inundated at $Q_p$ >= 1,000 m³ s$^{-1}$.

To check the plausibility of our model results, we mapped the uppermost reach of the Seti Khola between Karuwa and Kharapani (Fig. 1), where the 2012 flood caused most damage. We used high-resolution Google Earth satellite imagery from December 2011 (pre-flood) and June 2012 (post-flood) to map man-made structures, including houses, huts, and temples. We also recorded the extent of sediment deposited during the May 2012 flood along this 8.4-km long reach from orthorectified 5-m resolution RapidEye imagery of October 18, 2012. We compared the simulated flood areas with the mapped extent of flood sediments and the mapped changes in buildings.

## 4 Results

### 4.1 Impacts of the 2012 flood and flood scenarios

Our mapping of the May 2012 flood impacts along the Seti Khola show that nearly 30% of the 145 man-made structures visible in satellite images before the flood were undetectable after. We estimate a loss in built-up area of 14%. In the most heavily affected reach near Kharapani, only five of the 29 buildings remained, while 60% of built-up area was lost (Fig. 3). The best agreement between modelled inundation extent and observed flood sediment in this reach is for a simulated $Q_p$ of 3,700 m³ s$^{-1}$, which underestimates inundated areas by 14%.

Our scenario-based simulations reveal a spatially variable downstream pattern of potentially inundated areas along the Seti Khola (Fig. 4). In the uppermost reach north of the Mardi Khola confluence, simulated flood extents are very similar for all scenarios but funnel out further downstream where the May 2012 flood deposited gravel sheets. Gravel bars near or below Kharapani are currently exploited for gravel mining and likely to be flooded in all scenarios; local flow depth be as much as 14 m for peak discharge $Q_p$ of 10,000 m³ s$^{-1}$ (Fig. 4).

Between the Mardi Khola and Kali Khola confluences, inundation would largely affect a number of informal settlements and infrastructure as well as the Seti dam, given that the lowermost river terrace would be extensively flooded for $Q_p$ >3,000 m³ s$^{-1}$. One of these informal settlements at Yamdi would be inundated in all scenarios (Fig. 5), together with a major road

connecting Pokhara with north-western Nepal. For another informal settlement at Kaseri, all flood scenarios indicate extensive flooding with a mean flow depth of 23 m and a mean flow velocity of 1.3 m s$^{-1}$ for $Q_p$ = 10,000 m³ s$^{-1}$ (Fig. 6).

Further downstream, Ramghat is another informal settlement and also a site of religious importance that could be affected by up to 32-m deep flows in the $Q_p$ = 10,000 m³ s$^{-1}$ scenario. Flood-water levels could reach the edge of the uppermost terrace if $Q_p$ >7,000 m³ s$^{-1}$ and thus affect the surrounding dense urban areas (Fig. 4).

The less confined, meandering channel downstream of the Phusre Khola confluence has the largest variations in inundation areas in the Pokhara valley, and point bars mined for gravel or used for crop production are likely to be flooded at $Q_p$ >6,000 m³ s$^{-1}$. We observe pronounced backwater flooding in at least three tributaries of the Seti Khola in all $Q_p$ scenarios, likely inundating the lowest reaches of these tributaries for several hundred metres for $Q_p$ >6,000 m³ s$^{-1}$ (Fig. 4). Our results also show a patch of inundation in central Pokhara, some 1.5 km away from the Seti Khola and close to the Phirke Khola tributary (Fig. 4).

## 4.2 Potential future flood impacts

In 2020, 41% of the study area was covered by developed land, while mixed forest covered 27% mostly in the northern and southern parts (Fig. 7). Our simulations indicate that grassland, forests and barren classes (including heavily mined gravel bars) would be flooded widely in all $Q_p$ scenarios (Table 2, Fig. 8). Some 0.5 km² or 2.3% of the total crop area could be submerged for $Q_p$ = 10,000 m³ s$^{-1}$. Of all developed LC classes, 0.3 km² or 0.6% would be affected by a peak discharge $Q_p$ of 1,000 m³ s$^{-1}$, compared to 2.8% (1.5 km²) for a peak discharge ten times higher. Low developed areas appear to be affected most extensively in all scenarios (Fig. 9). For the worst $Q_p$ scenario, some 0.6 km² of both low (2%) and medium developed (4.1%) area would be flooded, as opposed to 0.09 km² (1%) and 0.25 km² (2.8%) for the densest urban areas. Thus, most of the affected areas of the "developed – high" and "developed – open" classes are rated as hazard class 10 (Table 2, Fig. 9). Only airport areas seem completely devoid of flooding under the scenarios considered here.

Our analysis of the 2021 OSM route network shows that 4% of its length is prone to flooding, with 0.7% being in the highest hazard class 10 (Table 3, Fig. 10). Most roads and paths with HC >7 run along the upper Seti Khola upper reach, upstream of Kaseri.

As of September 2021, OSM users have mapped a total of 62930 buildings in Pokhara and 2.6% of these buildings and 2.5% of the total built-up area are prone to flooding scenarios (Table 3, Fig. 11). A peak discharge of 10,000 m³ s$^{-1}$ could affect 282 buildings (0.5%), which translates to 0.3% (0.02 km²) of the total built-up area falling into hazard class 10. Many of the buildings categorized into the higher hazard classes (HC >7) are in Pokhara's north-western urban areas, especially in the Yamdi and Kaseri settlements.

### 4.3 Informal settlement dynamics

The two informal settlements at Kaseri and Yamdi developed rapidly since 2008 (Table 4, Fig. 12). We find that Kaseri had expanded from a low terrace in 2008 towards the floodplain by 2012, thus more than doubling its built-up area. This growth continued until at least 2021, covering more than a fifth of the overbank area. The built-up area at Yamdi covered a much lower proportion of overbank area in 2008 (0.4%). Yet, growth was more rapid since, and the built- up area more than tripled between 2008 and 2012 and increased by a factor of six from 2012 to 2021. Comparing November 2012 and November 2021, we also observed a significant expansion of sand- and gravel mining in this particular reach.

### 5 Discussion

### 5.1 Inundation modelling

Our scenario-based models of potential flood impacts in Pokhara highlight that especially recent and rapidly growing informal settlements close to the active channel have a high likelihood of being inundated. Before we discuss the implications, we comment on the applicability of our methods. HEC-RAS has been widely applied to model sudden floods (Cenderelli and Wohl, 2003; Klimeš et al., 2014; Wang et al., 2018), albeit at the cost of simplifying flood dynamics in terms of sediment content and channel-bed stability. Dynamic entrainment of sediments by bed erosion may alter flow rheology and runout (Westoby et al., 2014). Exact channel geometries are inaccessible in the high-alpine headwaters of the Seti Khola, thus curtailing the use of physically more advanced flow models. Although these could handle erosion and sedimentation dynamics in the propagation of sediment-laden flows, many of the required input parameters remain unconstrained (Cesca and D'Agostino, 2008; Westoby et al., 2014; Zhang and Liu, 2015). While the geomorphic setting caters to numerous potential flood-water sources and generating mechanisms, it rarely conserves evidence of initial breach characteristics. Yet these unknown parameters are essential for the empirical estimation of hydrographs that are required in models of unsteady flow; hence, we restricted our scenario simulations to one-dimensional steady flow.

The accuracy of our results hinges on the accuracy of river cross-sections and the estimates of channel and overbank roughness (Manning's $n$; Westoby et al., 2014; Wohl, 1998). Previous studies of HEC-RAS for sudden floods have used mostly coarser digital elevation data than the 5-m ALOS DEM we used here (Mergili et al., 2011; Somos-Valenzuela et al., 2014; Wang et al., 2018; Zhang and Liu, 2015). The stereo satellite imagery forming the basis for this DEM was acquired between 2006 and 2011 and excludes channel changes by the May 2012 flood (Gurung et al., 2021). We minimised potential resulting effects on our models by manually adjusting cross-sections with our additional field-surveyed elevation data. We tested the sensitivity of our HEC-RAS simulations related to the choice of Manning's $n$ by comparing the results of two $Q_p = 10,000$ m³ s⁻¹ scenario models, one with spatially varying and one with a constant Manning's $n$. Variations between these two models are minimal such that a spatially varied Manning's $n$ slightly improves the local accuracy of simulations. Yet these local improvements

might be crucial in assessing flood hazard for the informal settlements near the Seti Khola. If the accuracy of our simulations of the May 2012 flood is anything to go by, we surmise that our inundation scenarios are potential underestimates.

Our HEC-RAS simulations also show apparent flooding in places that are likely artefacts of the poorly resolved geometry of several narrow gorges and some subsurface drainage (Fig. 4). Sinkholes and caves in the Ghachok Formation may route some of the discharge of the Seti Khola below the surface, especially in urban areas (Fort, 2010; Rimal et al., 2015; Stolle et al., 2019). Modelling the groundwater flow in these potential karst structures is beyond the scope of this study and would require hydrological details that remain largely unresolved. We thus modelled flood flows for an idealized gorge geometry informed by DEM and field data. Hence, model artefacts potentially occur at cross-sections where this idealised gorge geometry fails to capture or underestimates subsurface flow. Thus, interpretations of the resulting highly localised inundation areas should be handled with care.

## 5.2 Flood scenarios and their potential impact

Our results offer a first comprehensive set of non-meteorological flood scenarios along the Seti Khola, and thus expand on previous studies of a more qualitative (Rimal et al., 2015, 2018) or local focus (Gurung et al., 2021; Thapa et al., 2022). By intersecting modelled inundation extents with spatial data on individual buildings and the route network, we are able to outline relative hazard zones on the assumption that smaller flood magnitudes are more frequent than larger ones. We show that zones with higher relative flood hazard (HC >7) are mostly along the Seti Khola's upper reach. Despite the May 2012 flood's intense damage to infrastructure in this area, a number of infrastructure projects have been developed close to the Seti Khola, including several road bridges and a run-of-the-river hydroelectric station (Gurung et al., 2021).

Further downstream, our mapping of the recent spread of informal settlements along the Seti Khola at Kaseri and Yamdi between 2008 and 2021 substantiates residence interviews about this rapid growth (Gurung et al., 2021). We found that unregulated sand and gravel extraction at Yamdi has, like in many other Nepalese rivers, increased in past decades, but has especially accelerated after large amount of sediments were deposited by the May 2012 flood (Dahal et al., 2012; Fort et al., 2018). Our results illustrate the coupling of intense flood sediment deposition and built-up area expansion along the Seti Khola's mid reaches, where more people were and still are attracted by this emerging opportunity of income. Although interviews with gravel and sand miners have shown that they are aware of this risk (Henzmann, 2020), they successively move into these flood-prone locations and raise their vulnerability to floods. Gravel mining of this scale is likely to further modulate the roughness of the active channel, while enhancing bank erosion and cliff collapse, thus propagating the flood hazards also to the densely populated river terraces further up (Fort et al., 2018; Kondolf, 1994).

One key finding of our simulations is that hydraulic ponding upstream of gorges and backwater flooding at tributaries occurs in all scenarios, leading to high local flow depth and reduced flow velocities. The Seti dam and Kaseri informal settlement appear to be particularly prone to hydraulic ponding, which might cause severe flooding there. Locally sustained flood peaks might cause spatially much more diverse flood hazard along this reach. While our models simulate clear-water flow, hydraulic ponding is likely exacerbated by entrained sediment and debris that may further clog or block the gorges (Thapa et al., 2022).

Hydraulic ponding is a common process during sudden floods in bedrock or resistant-boundary channels and were reported for Holocene jökulhlaups in Iceland (Carrivick, 2006, 2007) and historic outbursts of moraine-dammed lakes in British Columbia (Kershaw et al., 2005). This importance of hydraulic ponding complements the findings of previous studies concerned with estimating flood frequency at several sites (Basnet and Acharya, 2019; Gurung et al., 2021). Estimates of the 100-year flood based on rainfall data are $Q_p$ = 2,336 m³ s⁻¹ (Basnet and Acharya, 2019) and $Q_p$ = 2,423 m³ s⁻¹ (Gurung et al., 295 2021). These are consistent with the lower range of non-meteorological flood scenarios and offer comparable flood limits in the Ramghat (Basnet and Acharya, 2019) and Kharapani areas (Gurung et al., 2021).

### 5.3 Recommendations

No early-warning strategy was in place during the May 2012 flood, and several authors surmised that a chance warning presumably prevented a higher death toll in the north-western outskirts of Pokhara (Gurung et al., 2015; Kargel et al., 2013). 300 In the flood's immediate aftermath, public flood risk awareness and preparation trainings as well as an early-warning system were implemented – including a water-level sensor in the Seti Khola just above the furthest upstream settlement (Gurung et al., 2015). However, recent interviews with local residents showed that a lack of maintenance has rendered the early warning system inoperable in the past years (Henzmann, 2020). Thapa et al. (2022) pointed out, that existing evacuation routes from Yamdi and Kaseri settlements towards higher ground are inadequate and outlined several structural (e.g. river embankment) 305 and non-structural (e.g. evacuation rehearsal drills, population relocation) mitigation measures. The authors proposed to install weather and hydrological stations and to transmit early warnings via mobile phone. However, their flood response strategy strictly focuses on urban areas in Pokhara's north-west. We argue, that a regularly maintained warning system might want to provide full coverage of settlements along the Seti Khola's course through the Pokhara valley, including rural settlements and exposed infrastructure in the northern valley. The May 2012 flood also demonstrated that such flash floods can travel fast in 310 the steep headwaters of the Seti Khola (Kargel et al., 2013; Oi et al., 2014). Stream gauges and comparable monitoring stations may need to be located further upstream than presently implemented, ideally close to the outlet of Sabche Cirque to maximise warning times for downstream communities. Complementing Thapa et al.'s (2022) proposal for early-warning strategies in the downstream reaches, we argue that a special focus should be on monitoring water and sediment dynamics in the Seti Khola's headwaters. Optical and SAR satellite data acquired at short repeat rates, for example, via the Sentinel and Planet platforms, 315 might further assist early warning, as ground movement and deformation of the cirque walls, surge phases of the Sabche glacier, and lake formation might be tell-tale warning signs (Grebby et al., 2021; Hermle et al., 2021; Kirschbaum et al., 2019; Quincey et al., 2005). Following Zhang and Wang (2022) on their mitigation efforts of outburst hazard from Lake Cirenmaco, China, field monitoring could include the real-time transmission of optical and thermal data, captured by 360° cameras, to a round-the-clock-operating data processing centre. Yet, the potential cloudiness of this area together with increasing pollution 320 may constrain visual monitoring. Downstream, however, urban planning may have different priority for decision makers (Gurung et al., 2021). Instead of relocating people from vulnerable informal settlements to safer places, the government and

local authorities may have encouraged people to live near riverbanks or on floodplains by providing them with basic amenities (drinking water, electricity, access road) and land owner certificates (Gurung et al., 2021).

## 6 Conclusions

The Seti Khola is the lifeline of Nepal's second largest city, but also a river prone to sudden floods, given that it traverses one of the steepest topographic gradients in the Higher Himalayas. We provide the first comprehensive assessment of potential discharge scenarios and intersect hydrodynamically modelled inundation extents with land-cover data. Our simulations demonstrate the high spatial variability of potential flood impacts in the Pokhara valley. All model runs point to potential hydraulic ponding with high flow depths and low flow velocities above deeply incised gorges in urban areas.

We find that even a moderate non-meteorological flood scenario with peak discharge well within the ranges of monsoonal floods ($Q_p = 1,000$ m³ s$^{-1}$), could inundate some 0.6% (0.3 km²) of the developed area and 3.2% of cropland and grassland. A larger flood involving a peak discharge ten times higher would flood 2.8% (1.5 km²) of the Pokhara valley's developed areas and 9.8% of its agricultural areas. OSM data of built-up area in Pokhara's urban areas show, that relative inundation hazard is highest in the city's north-western outskirts, where a rapidly growing number of informal buildings linked to gravel mining is

has moved close to the Seti Khola. These sites of extensive unregulated sand and gravel mining would be extensively inundated in all our discharge scenarios – putting workers and informal settlement dwellers at risk. Built-up area in both Kaseri and Yamdi has grown intensively in the past decade: While built-up area in Kaseri, Pokhara's oldest established informal settlement, tripled between 2008 and 2021, built-up area rapidly increased by a factor of 20 in the recently forming settlements at Yamdi. Since 2012, a sixfold increase in built-up area at the latter is accompanied by an intense expansion of gravel and

sand mining activities. Several roads that are crucial for the supply reliability of rural communities are also prone to more frequent flooding as these roads run parallel to the Seti Khola on its lower alluvial terraces.

We conclude that urban planning and risk mitigation strategies in the Pokhara valley might wish to consider the hazard posed by sudden, non-meteorological floods in more detail, given the accumulating evidence of repeated historic and prehistoric events. Potential risk reduction measures may include in-field and remote monitoring of the Seti Khola's headwaters as well

as early-warning strategies, including a statutorily determined chain of warning as well as public awareness training. Such training could be aided by considering scenario-based limits of inundation for different flood sources.

## Appendix

Figure A1

FigureA2

## Data availability

All field data was published by Fischer et al. (2022) in the PANGAEA Data Publisher for Earth & Environmental Science (https://www.pangaea.de/) and is freely available under https://doi.org/10.1594/PANGAEA.941540. ALOS WORLD 3D Topographic Data was provided by the Remote Sensing Technology Center of Japan (©NTT DATA, RESTEC/ Included©JAXA). PlanetScope and RapidEye satellite imagery was freely provided through the Education and Research Program of Planet Lab Inc. (https://www.planet.com/explorer). Data from the OpenStreetMap project are freely available via https://www.openstreetmap.org/ and https://overpass-turbo.eu/. We further used the freeware HEC-RAS 5.0.7, which is available under https://www.hec.usace.army.mil/software/hec-ras/, and Google Earth Pro (https://www.google.com/earth/versions/#earth-pro).

## Author contributions

Me.F., A.W., O.K., and S.R. conceptualised the study. The field data were collected by Me.F. and O.K. while Me.F. and J.B. curated field and additional data. All data were processed and visualised by Me.F. Me.F. prepared the original manuscript, which was reviewed and edited by A.W., S.R., Mo.F. and O.W.

## Competing interests

The authors declare that they have no conflict of interest.

## Acknowledgements

This research was supported by the Deutsche Forschungsgemeinschaft (DFG) via the graduate research training group NatRiskChange at the University of Potsdam (https://www.natriskchange.de). The authors thank Georg Veh, Elisabeth Schönfeldt, and Narayan Gurung for their support during fieldwork and Natalie Lützow for data support with HEC-RAS. Adam Emmer and an anonymous referee provided helpful reviews of an earlier manuscript version.

## Financial support

This study was financed by the Deutsche Forschungsgemeinschaft (GRK 2043/1 and 2043/2).

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

**Tables**

**Table 1: Manually mapped land cover (LC) and land use (LU) classes. LC definition follows the USGS National Land Cover Database 2019 (NLCD2019; Multi-Resolution Land Characteristics (MRLC) Consortium, 2019), complemented by two LU classes.**

| Class names | | Notes | Classsification base |
|---|---|---|---|
| Land Cover | Land Use | | |
| developed, open | | <20% covered by impervious materials | NLCD2019 |
| developed - low (intensity) | | 20% to 49% covered by impervious materials | NLCD2019 |
| developed - medium (intensity) | | 50% to 79% covered by impervious materials | NLCD2019 |
| developed – high (intensity) | | >80% covered by impervious materials | NLCD2019 |
| | airport | | this study |
| (open) water | | | NLCD2019 |
| barren (land) | sand and gravel mining | | NLCD2019 and this study |
| grassland(/herbaceous), | | might include grazing of livestock | NLCD2019 |
| (cultivated) crops | | | NLCD2019 |
| shrub (scrubs) | | | NLCD2019 |
| (mixed) forest | | | NLCD2019 |

**Table 2: Hazard matrix listing the area (in ha) of each LULC class located within the respective hazard class (HC)**

| Hazard Class | Developed - high | Developed - medium | Developed – low | Developed - open | Airport | Barren | Grassland | Crops | Shrub | Forest |
|---|---|---|---|---|---|---|---|---|---|---|
| HC 10 ($Q_p >= 1{,}000$ m³ s⁻¹) | 8.8 | 5.0 | 14.7 | 4.8 | 0 | 78.6 | 89.1 | 4.9 | 26.2 | 59.2 |
| HC 9 ($Q_p >= 2{,}000$ m³ s⁻¹) | 4.0 | 2.7 | 6.7 | 2.2 | 0 | 8.7 | 27.6 | 3.0 | 7.5 | 11.7 |
| HC 8 ($Q_p >= 3{,}000$ m³ s⁻¹) | 2.4 | 1.8 | 6.4 | 0.3 | 0 | 4.9 | 20.3 | 3.4 | 6.5 | 8.8 |
| HC 7 ($Q_p >= 4{,}000$ m³ s⁻¹) | 1.4 | 4.3 | 6.3 | 0.4 | 0 | 2.1 | 20.6 | 3.4 | 7.9 | 7.6 |
| HC 6 ($Q_p >= 5{,}000$ m³ s⁻¹) | 1.1 | 5.0 | 4.5 | 0.3 | 0 | 1.6 | 18.4 | 4.6 | 5.3 | 8.6 |
| HC 5 ($Q_p >= 6{,}000$ m³ s⁻¹) | 1.1 | 3.0 | 3.2 | 0.09 | 0 | 1.8 | 16.4 | 4.6 | 4.9 | 8.3 |
| HC 4 ($Q_p >= 7{,}000$ m³ s⁻¹) | 1.1 | 4.6 | 3.3 | 0.01 | 0 | 1.2 | 15.9 | 5.2 | 4.6 | 6.5 |
| HC 3 ($Q_p >= 8{,}000$ m³ s⁻¹) | 1.2 | 4.1 | 3.8 | 0.01 | 0 | 1.0 | 13.8 | 5.0 | 5.0 | 5.3 |
| HC 2 ($Q_p >= 9{,}000$ m³ s⁻¹) | 1.6 | 12.9 | 4.7 | 0.01 | 0 | 0.7 | 11.2 | 6.7 | 4.2 | 4.9 |
| HC 1 ($Q_p >= 10{,}000$ m³ s⁻¹) | 2.1 | 15.3 | 8.2 | 0.01 | 0 | 0.4 | 10.1 | 8.3 | 3.4 | 4.5 |
| No inundation | 861.0 | 1,379.9 | 3,080.3 | 62 | 205.7 | 19.3 | 605.7 | 2,077.3 | 703.7 | 3,507.4 |

**Table 3: Relative hazard classification of the Pokhara route network and buildings. Data from OpenStreetMap (September 2021).**

| Hazard Class | Route Network | | Buildings | | |
|---|---|---|---|---|---|
| | Length [km] | % of total | No. Buildings | Built-up area [ha] | % of total |
| HC 10 | 32.0 | 0.7 | 282 | 2.1 | 0.3 |
| HC 9 | 19.3 | 0.5 | 145 | 1.2 | 0.2 |
| HC 8 | 15.1 | 0.4 | 81 | 0.7 | 0.1 |
| HC 7 | 13.5 | 0.3 | 118 | 1.0 | 0.2 |
| HC 6 | 15.2 | 0.4 | 105 | 0.8 | 0.1 |
| HC 5 | 15.2 | 0.4 | 63 | 0.7 | 0.1 |
| HC 4 | 13.4 | 0.3 | 73 | 0.8 | 0.1 |
| HC 3 | 13.0 | 0.3 | 88 | 0.9 | 0.1 |
| HC 2 | 16.8 | 0.4 | 250 | 2.9 | 0.5 |
| HC 1 | 19.0 | 0.4 | 409 | 4.2 | 0.7 |
| No inundation | 4160.5 | 96 | 61316 | 607.4 | 97.5 |

**Table 4: Growth of built-up areas of informal settlements at Kaseri and Yamdi, Pokhara, 2008 - 2021.**

| | Kaseri | | Yamdi | |
|---|---|---|---|---|
| | Built-up area [ha] | % of overbank area | Built-up area [ha] | % of overbank area |
| Apr-2008 | 1.2 | 7.3 | 0.07 | 0.4 |
| Nov-2012 | 2.5 | 15.1 | 0.2 | 1.4 |
| Nov-2021 | 3.7 | 22.3 | 1.5 | 9.1 |

**Figures**

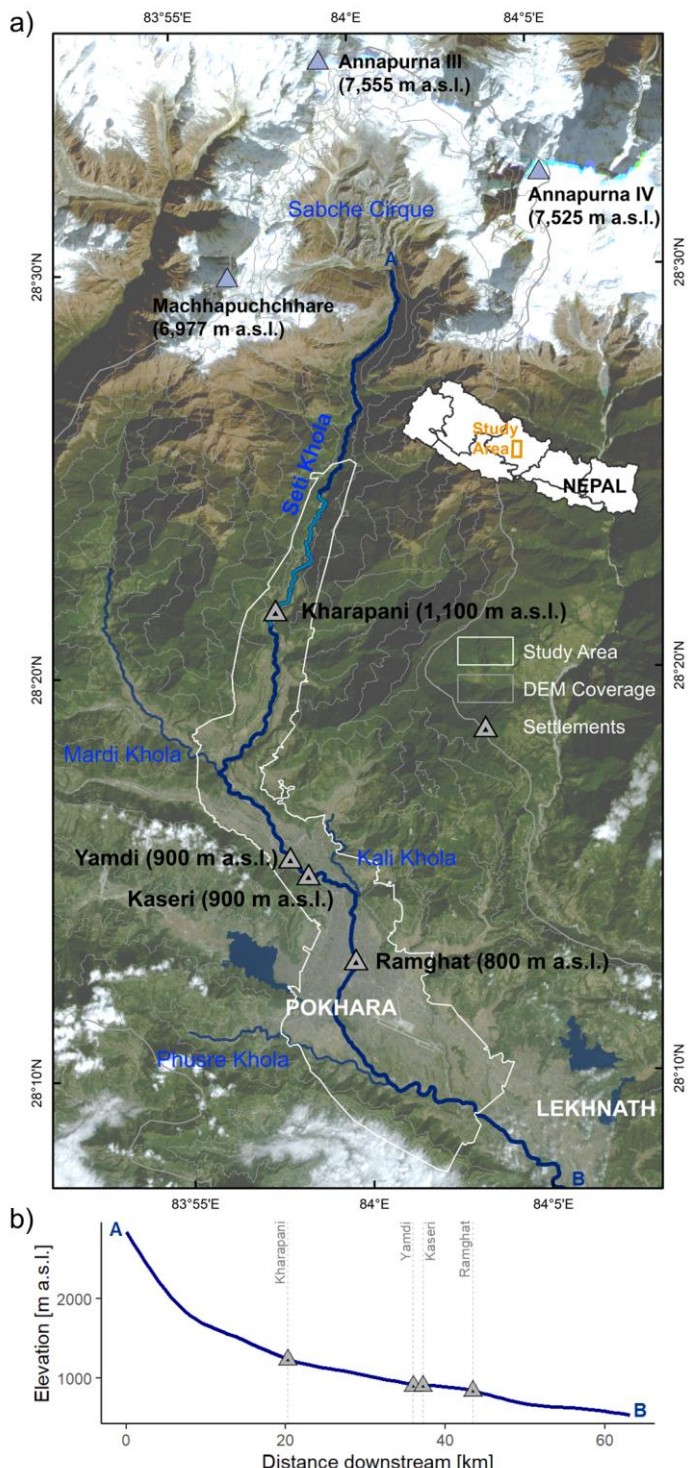

**Figure 1: The Pokhara valley and the Seti Khola in November 2021. a) Our HEC-RAS model domain and manual land cover mapping covers the study area (white polygon, Appendix Fig. A2). Most intense damages during the May 2012 flood occurred along the uppermost inhabited reach of the Seti Khola (light blue). Grey triangles mark settlements mentioned in this study. Image: PlanetScope (Planet Team, 2017); Contour lines (500 m spacing) are from AW3D DEM (©NTT DATA, RESTEC/ Included ©JAXA); Nepal administrative boundaries are from UN Office for the Coordination of Humanitarian Affairs - Field Information Services Section (OCHA FISS, 2020). b) Smoothed longitudinal profile of the Seti Khola from the Sabche Cirque (A) to the south-eastern periphery of Lekhnath (B).**

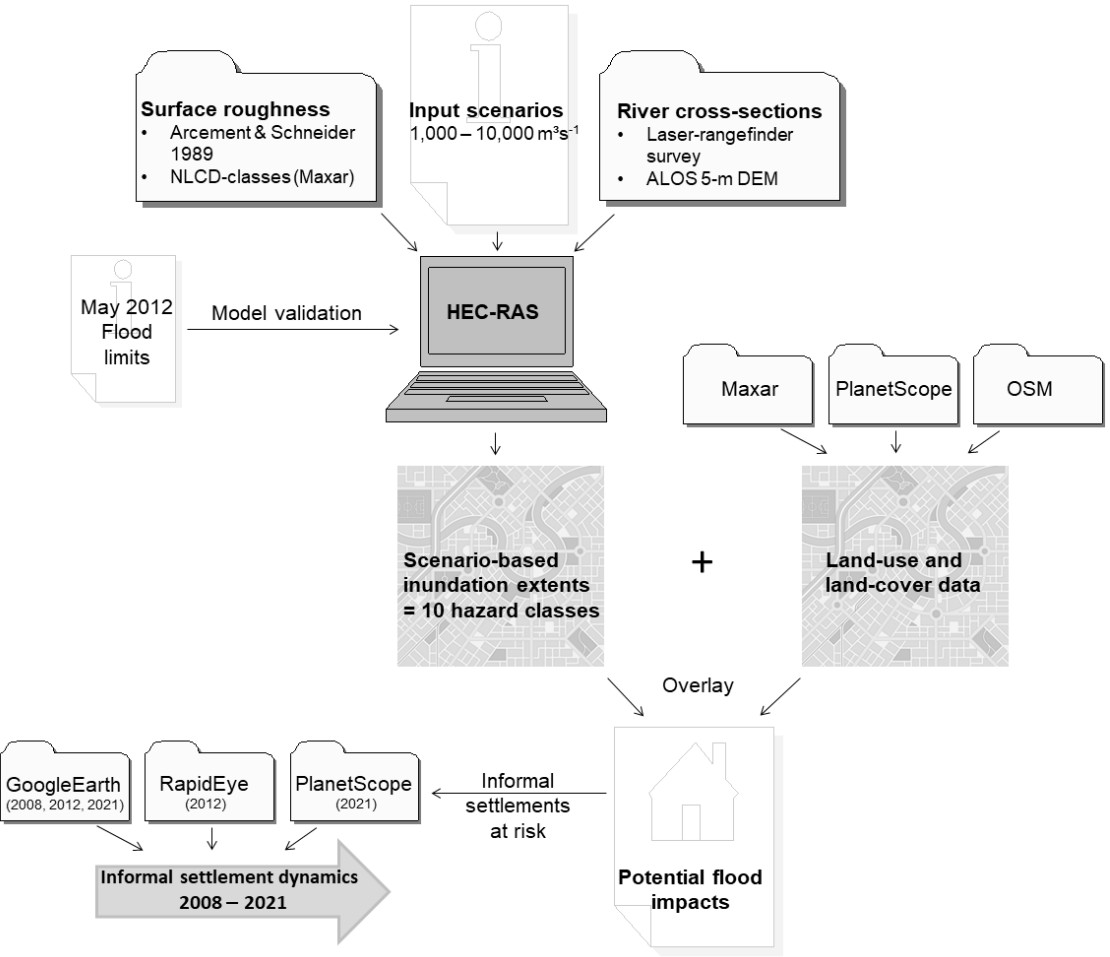

**Figure 2: Data sources and workflow of our semi-quantitative assessment of flood impacts in the Pokhara valley. See Appendix, Fig. A1 for model validation workflow.**

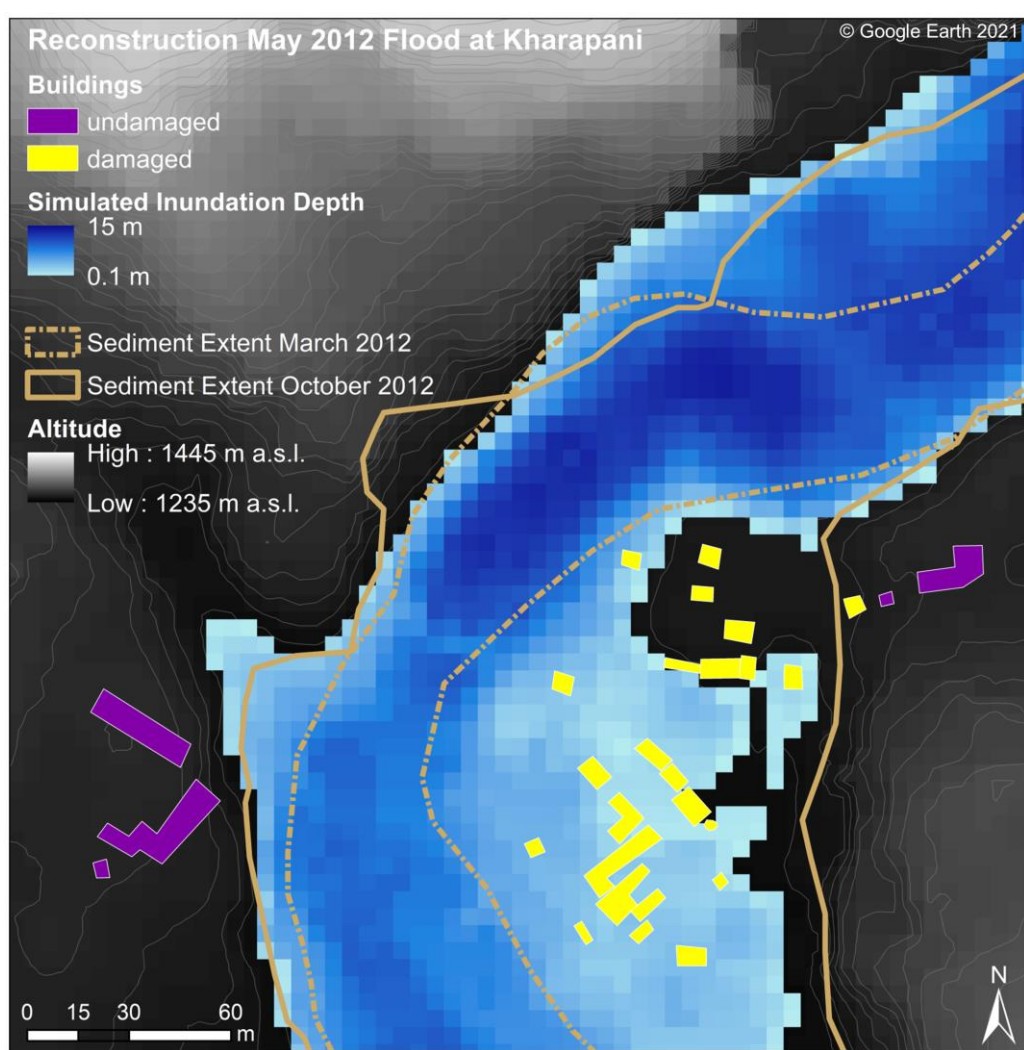

**Figure 3: Simulated flow depths of the May 2012 flood at Kharapani village for a flood peak of 3,700 m³ s⁻¹. Blue lines delimit patches of flood-derived sediment mapped from RapidEye imagery (October 2012). Building damage is based on comparing Google Earth imagery of December 2011 and June 2012.**

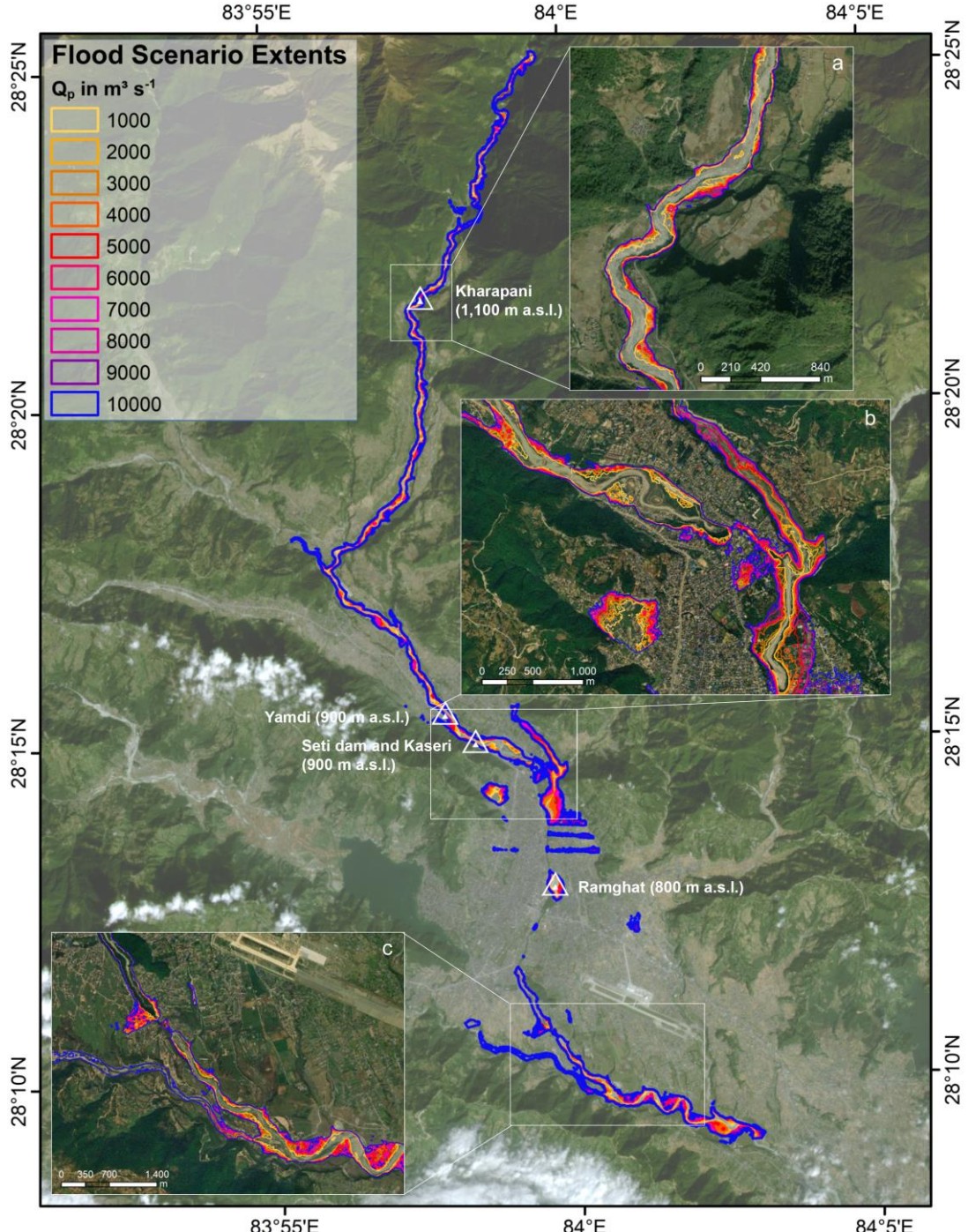

**Figure 4: Flood extents for HEC-RAS models of steady flow with $Q_p$ ranging from 1,000 m³ s⁻¹ to 10,000 m³ s⁻¹. Insets highlight results at Kharapani (a), Seti dam (b), and Phusre Khola confluence (c). Image: PlanetScope (13/11/2021) (Planet Team, 2017), inserts ESRI basemap Maxar imagery of 2020 (ESRI and Maxar Technologies, 2022).**

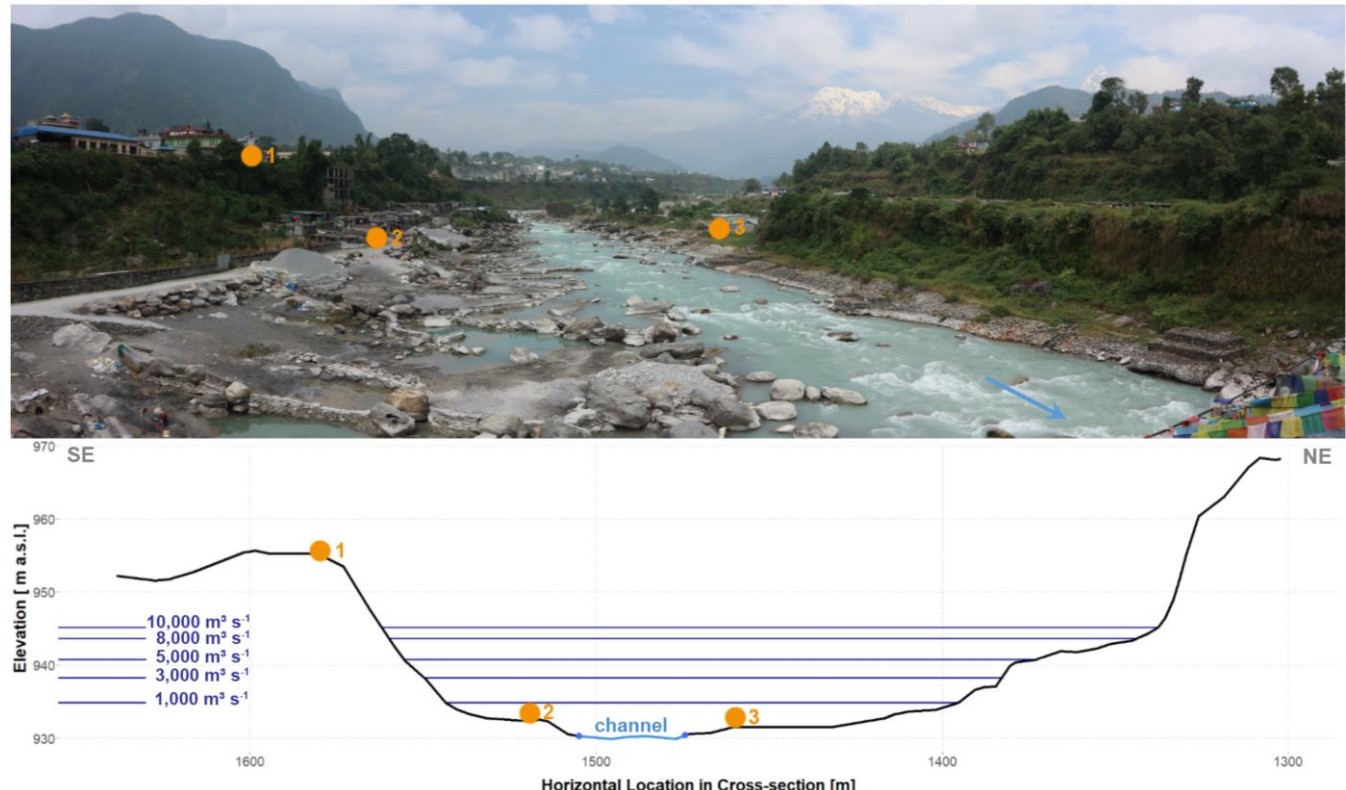

**Figure 5: Simulated water levels at an informal settlement and gravel mining site at Yamdi (Fig. 4) for five selected $Q_p$ scenarios. Buildings on overbank areas would be inundated in all modelled scenarios. Groynes and gravel heaps (in foreground) aid the detection of gravel mining activities from optical satellite images.**

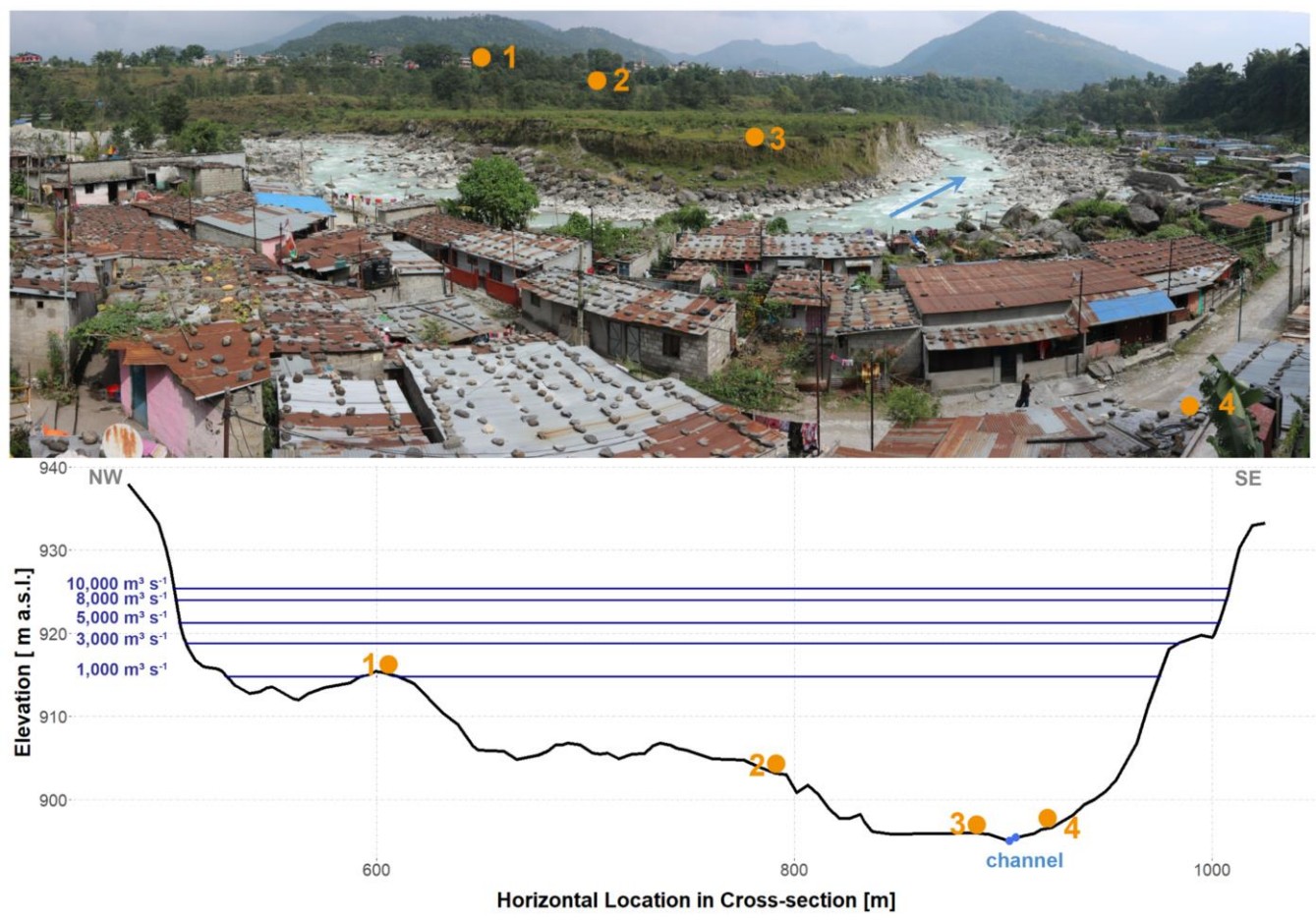

**Figure 6: Simulated water levels at the Kaseri informal settlement (Fig. 4) for five $Q_p$ scenarios. View from the edge of the settlement's core, which formed in the 1980's on a low terrace (Gurung et al., 2021), onto newer houses built close to the active channel bed of the Seti Khola.**

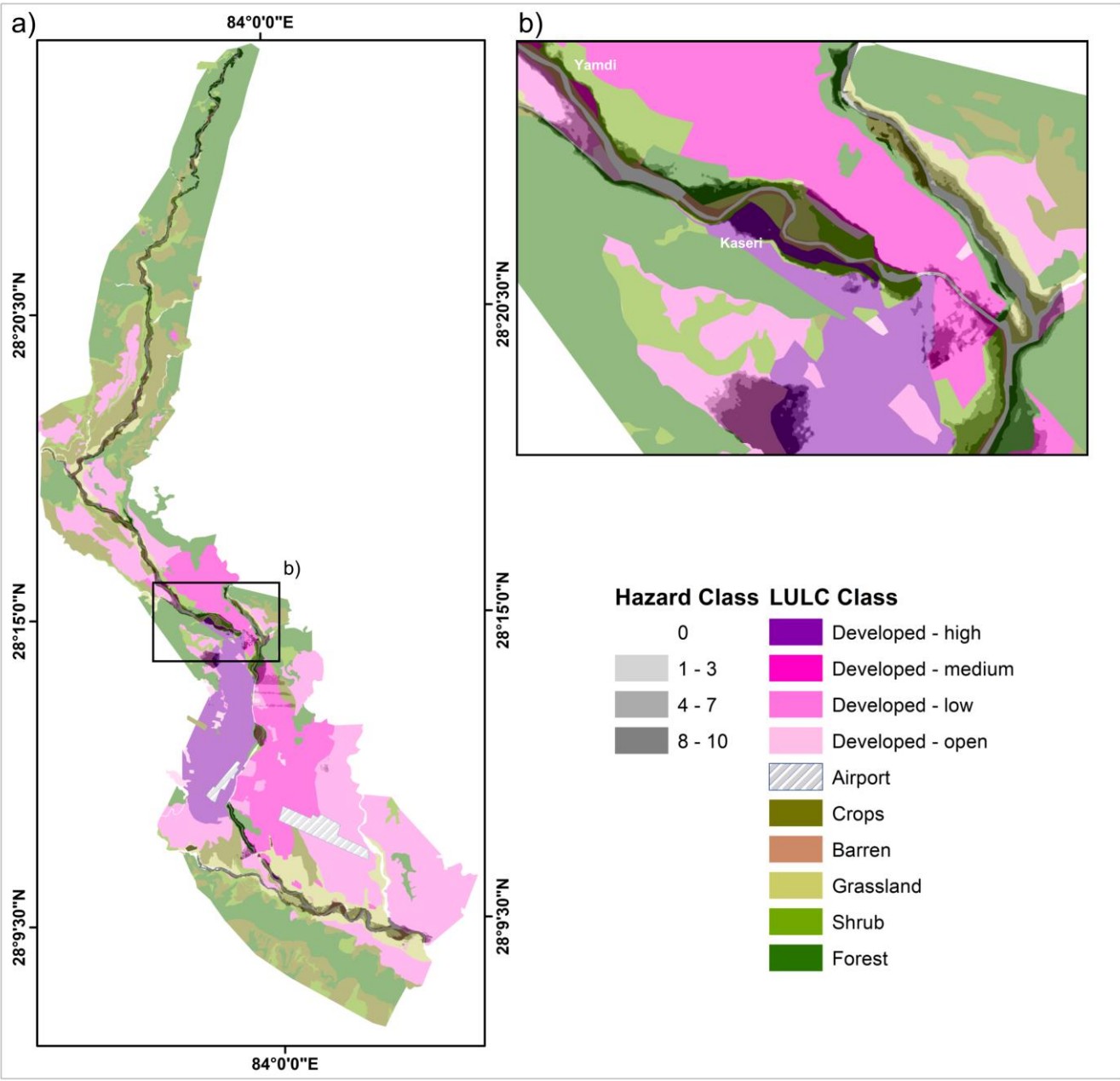

**Figure 7: Land use and land cover (LULC) map with flood hazard classes (a). Detail b) shows overlap of high hazard classes (HC 8 – 10) and densely populated areas (developed high to medium) at Yamdi and Kaseri informal settlements. LULC classes mapped from 2020 ESRI basemap Maxar imagery (ESRI and Maxar Technologies, 2022).**

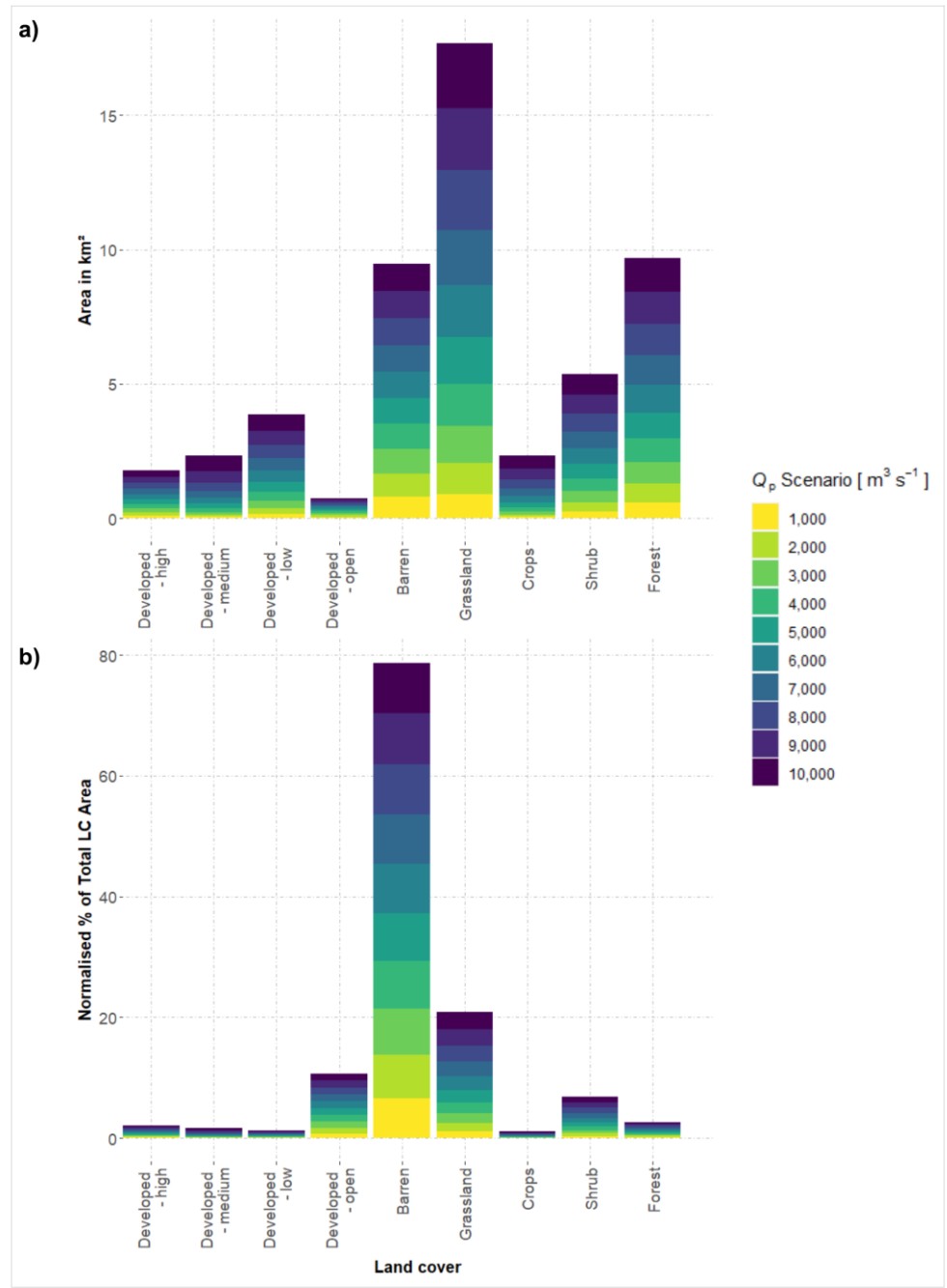

**Figure 8: Distribution of potentially flooded land cover (LC) classes in different $Q_p$ scenarios. a) Area of LC class affected; b) affected normalised percentage of total mapped LC class areas in our study area.**

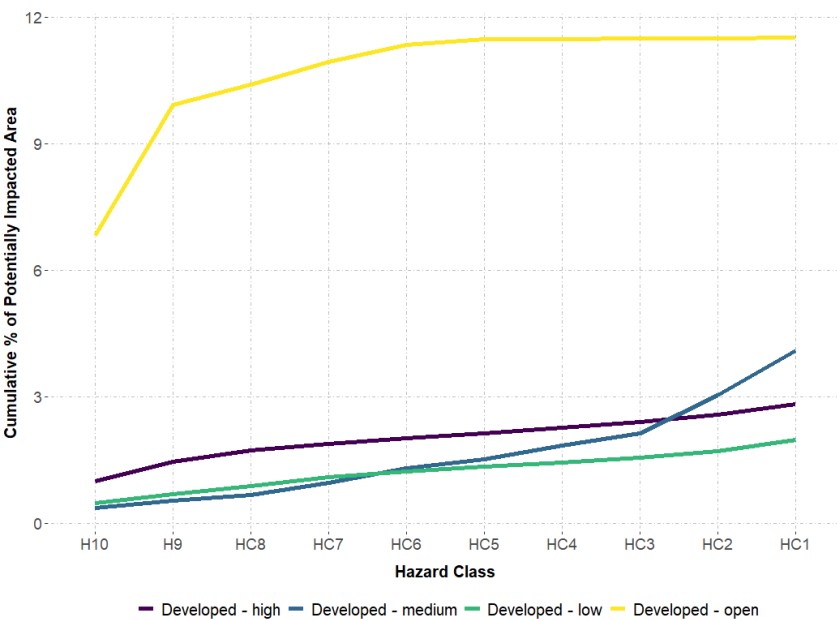

**Figure 9: Cumulative percentage of potentially flooded areas for each of the four developed LC classes.**

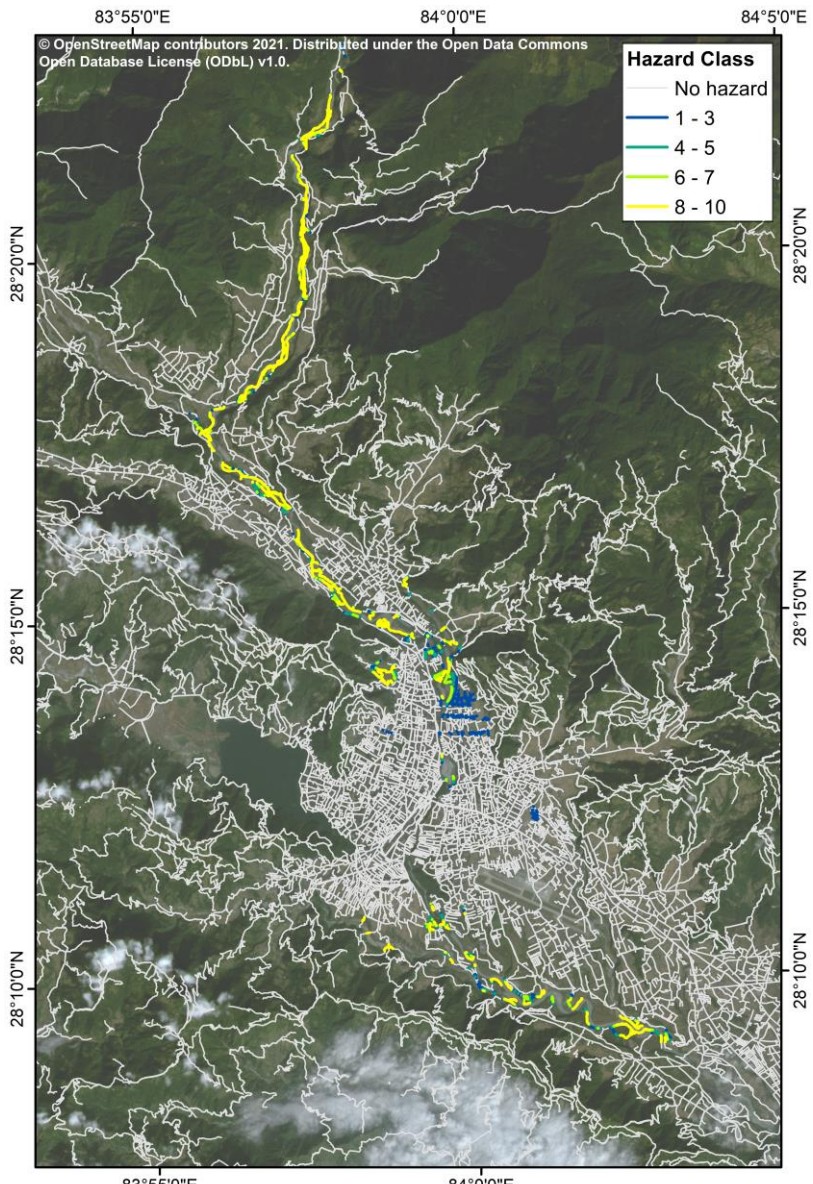

**Figure 10: Relative hazard classification of Pokhara's route network. Data from OpenStreetMap in September 2021 and draped on PlanetScope imagery (13/11/2021) (Planet Team, 2017).**

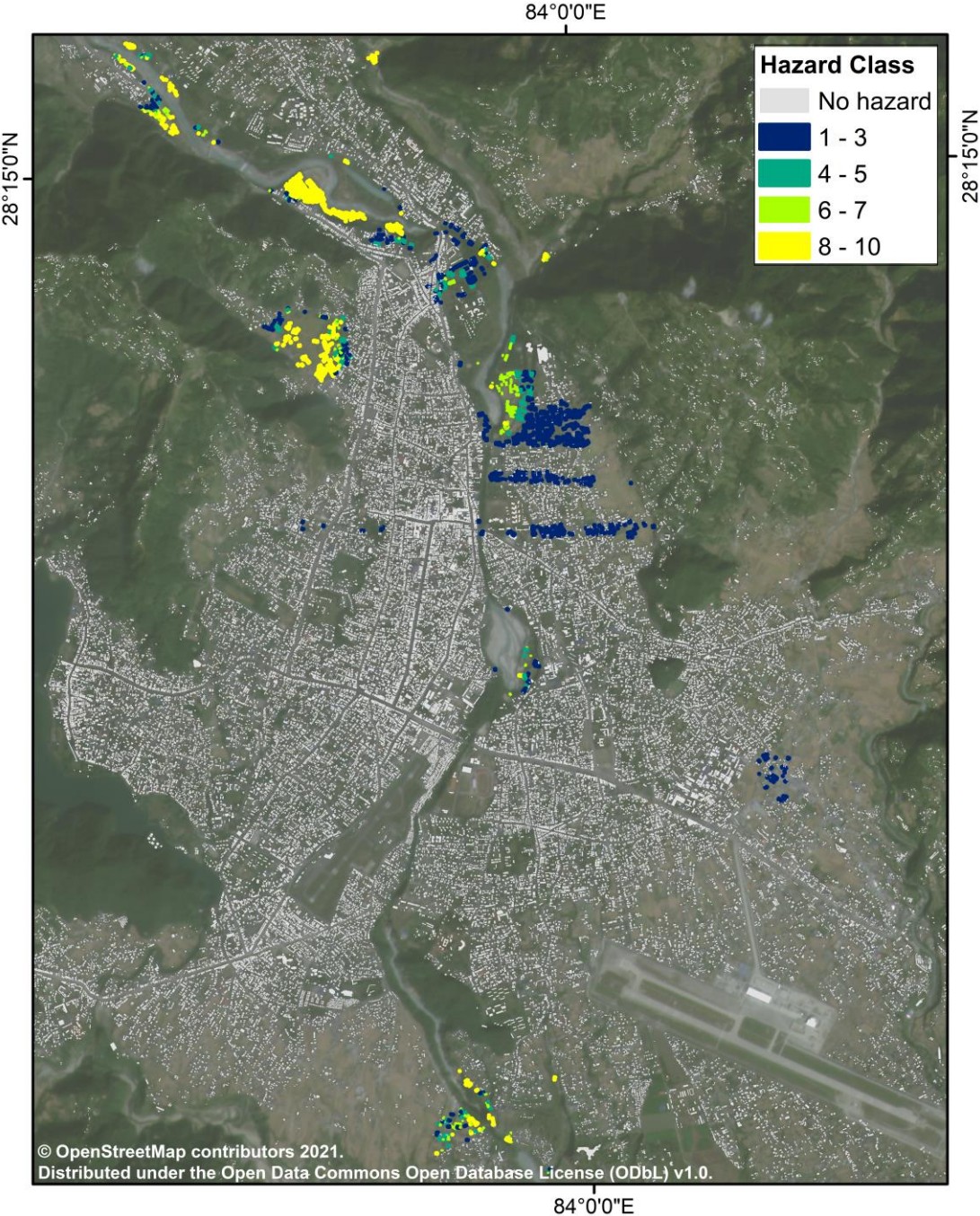

**Figure 11: Relative hazard classification of Pokhara's buildings. Data from OpenStreetMap (September 2021) draped on PlanetScope imagery (13/11/2021) (Planet Team, 2017).**

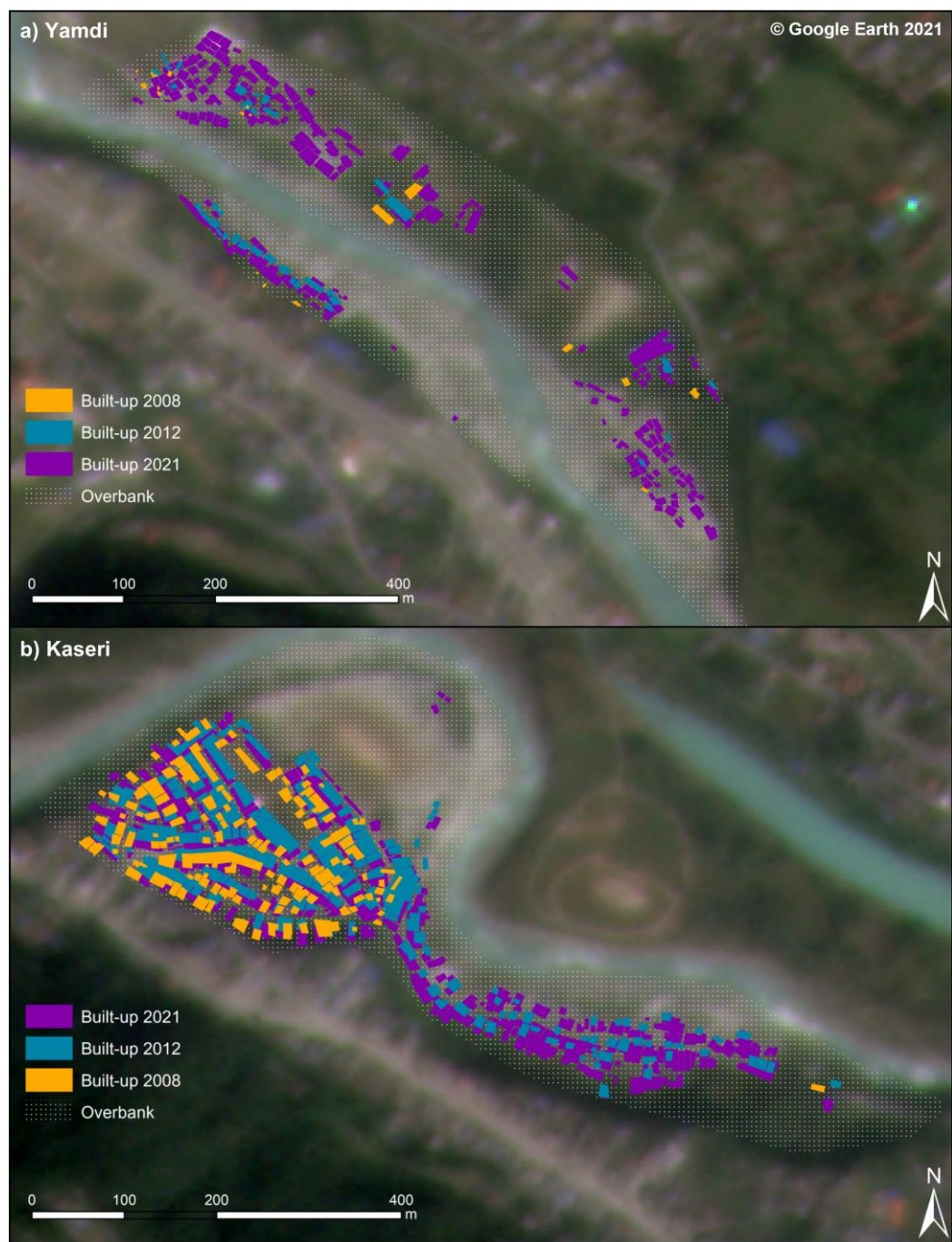

**Figure 12: Changes in built-up area between April 2008 and November 2021 at Yamdi (a) and Kaseri (b) informal settlements. Built-up area mapped from Google Earth and draped on Planet Scope imagery (13/11/2021) (Planet Team, 2017).**

**Figures Appendix**

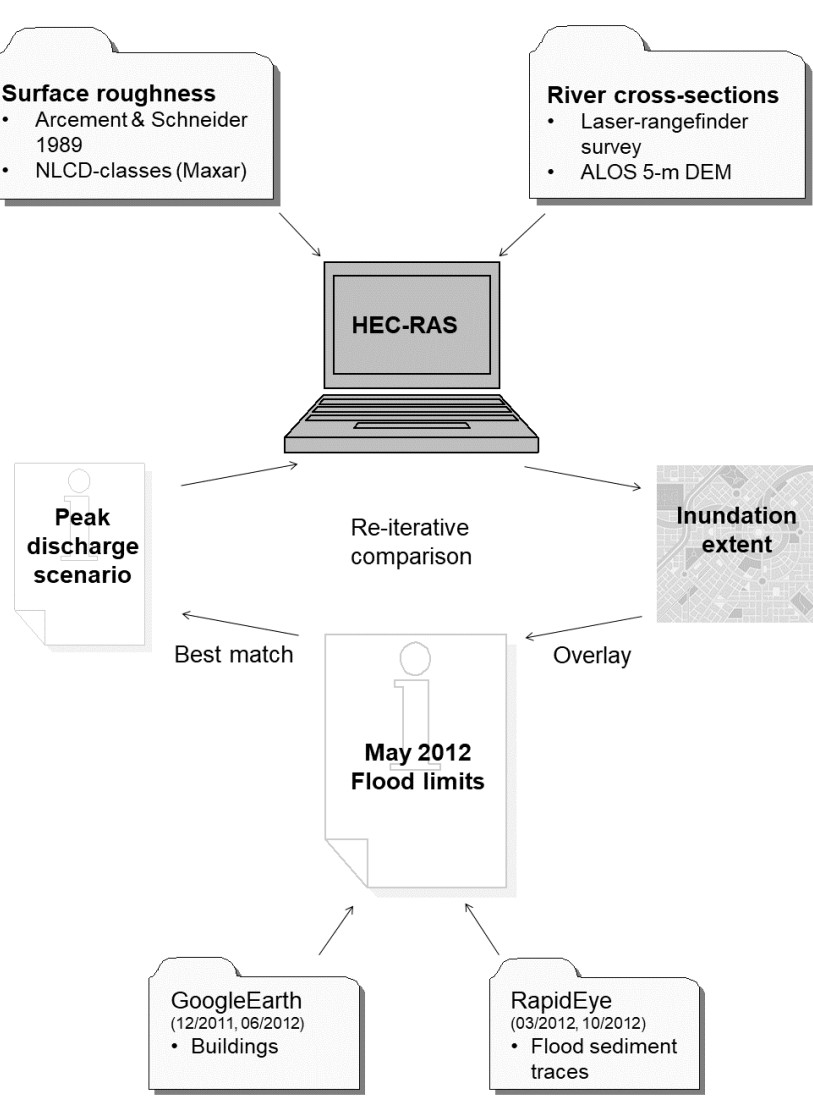

**Figure A1: Data sources and workflow of our model validation based on satellite-mapped sediment and damage traces of the May
2012 flood in the upper reach of the Seti Khola.**

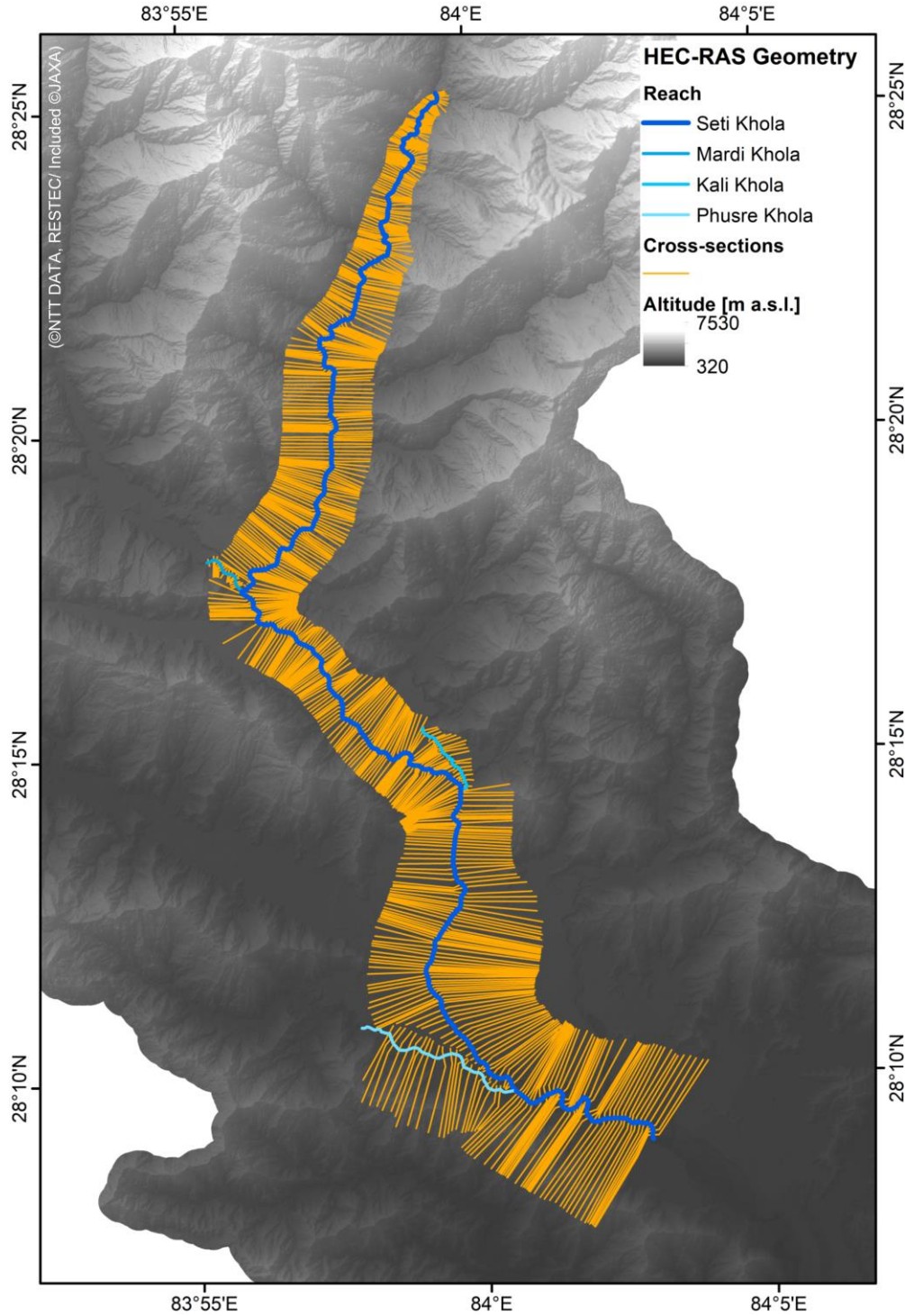

**Figure A2: Location of 572 cross-sections along the main (Seti Khola) and tributary reaches (Mardi, Kali, Phusre Khola) used as input to HEC-RAS models, plotted on top of the AW3D DEM (©NTT DATA, RESTEC/ Included ©JAXA).**