# Peer review of "Rare flood scenarios for a rapidly growing high-mountain city: Pokhara, Nepal"

_Natural Hazards and Earth System Sciences, 2022_

## Author Comment (AC1)

Reply to Referee Comment #1 by Adam Emmer

General Comments

*General Comment #1:*

> The authors repeatedly mention that the city is threatened by an outburst floods, but (nowadays) there are no lakes which could burst located upstream in the Sabche cirque. Therefore, the reader may wonder what could be the source of such flooding?

This is a good point, although the potential for impacts from outburst flood can change rapidly if lakes were to be impounded by landslides or glacier surges in the future. For more details, please refer to our reply to General Comment #3.

> I see a good argumentation with the 2012 flood, but it was also not an outburst flood according to the description provided by the authors (rather a highly mobile ice-rock avalanche which transformed into the hyper-concentrated flow; perhaps somewhat similar to the 2021 Chamoli disaster; 10.1126/science.abh4455).

We concur that the specific mechanism(s) of the 2012 "flood" remain(s) debated, if not elusive. Following this comment and also General Comment #4, we removed the term outburst flood when referring to the May 2012 flood and now draw some parallels to the 2021 Chamoli disaster (LL37ff of the original manuscript):

"Apart from annual monsoonal floods, this river has a history of rare, extreme floods. On May 5, 2012, a hyperconcentrated flow killed 72 persons and destroyed roads, bridges, and drinking water pipelines in the northern Pokhara valley(Gurung et al., 2015; Gurung et al.,2021). The exact sequence of events remains debated, but may have been initiated by rock-slope failures from the western flank of the Annapurna IV massif at 7,525 m a. s. l., observed by chance by a pilot (Hanisch et al., 2013; Kargel et al., 2013). Like in the 2021 Chamoli disaster (Shugar et al., 2021), a highly mobile ice-rock avalanche may have transformed into an hyperconcentrated flow that hit Kharapani village (1,100 m a.s.l.) some 23 km downstream just half an hour later, causing most of the damage and fatalities with an estimated peak discharge of 8,400 m³ s$^{-1}$ (Hanisch et al., 2013; Oi et al., 2014; SANDRP, 2014). Thanks to the pilot, a radio warning was issued, most likely preventing a higher death toll further downstream (Kargel et al., 2013)."

*General Comment #2:*

> Overall, the flood scenarios used are not well-justified and would benefit from better linkages to the past floods and potential future flood sources. For instance, why using 1,000 to 10,000 m3/s (1,000 m3/s step) and not e.g. 1,000 to 5,000 m3/s (500 m3/s step) if the 2012 flood corresponds to 3,700 m3/s in Kharapani. And it was likely less in Pokhara I guess – to put your flood scenarios in the context, could you also use HEC-RAS to estimate the 2012 peak in the city? If it was the largest recent flood, it could provide good guiding value for comparison and scenarios justification. Future

flood scenarios should be better connected to potential sources of these floods in my opinion.

Please refer to our reply to General Comment #3 concerning the linkages to future flood sources. We added more details about our choice of flood scenarios to section 3.1 (following LL99 of the original manuscript):

"We analyse potential flood impacts from physically plausible magnitudes of outburst floods along a 40-km long reach of the Seti Khola. Data to inform our range of scenarios come from two non-meteorological floods in the Pokhara valley, and have estimated peak discharges differing by orders of magnitude: while the May 2012 flood involved between 1,000 and 12,300 $m^3\ s^{-1}$ (Kargel et al., 2013; Oi et al., 2012; SANDRP, 2014), much larger Medieval floods may have involved 45,000 to 600,000 $m^3\ s^{-1}$ (Schwanghart et al., 2016), judging from geomorphic flood markers. Reported peak discharges from non-meteorological floods elsewhere in the Himalayas are 1,600 $m^3\ s^{-1}$ for the Dig Tsho GLOF of 1985 (Vuichard and Zimmermann, 1987), 8,000 to 14,000 $m^3\ s^{-1}$ for the 2021 Chamoli disaster (Pandey et al., 2022; Shugar et al., 2021), and provide further support for our range of historic flood scenarios."

*General Comment #3:*

Possible flood sources are very briefly touched in only one paragraph of discussion section 5.2, but I'm convinced it deserves more attention. My suggestion to the authors is to elaborate bit more on the potential source(s) of their otherwise virtual flood scenarios (sub-glacial outburst (?), glacial surge-indiced damming of the valley (?), outbursts of possibly landslide-dammed lakes (?), transformed ice-rock avalanche (?)). Could (some of) these processes lead to the impoundment / generation of enough water for 10,000 m3/s in the 30 km far city?

To motivate better our choice of scenarios, we move this paragraph from the Discussion to the Introduction (following LL44 of the original manuscript), and expanded it to:

"Even larger floods may have occurred in the Seti Khola in Medieval times, depositing much of the youngest sediment fill of the Pokhara valley during or shortly after large earthquakes, likely in the wake of major landslides or outbursts of glacier- and landslide-dammed lakes in the Annapurna Massif (Fort, 1987; Schwanghart et al., 2016; Stolle et al., 2017). Based on relict natural dams, Schwanghart et al. (2016) estimated that up to 1 $km^3$ of water could have been stored in the steep walled and sediment-filled Sabche Cirque some 35 km north of Pokhara city; outburst floods from this cirque could have released water at peak rates of up to 600,000 $m^3\ s^{-1}$. Several authors agree that the cirque might spawn large floods along the Seti Khola in the future (Fort, 2010; Grandin et al., 2012; Gurung et al., 2021; Kargel et al., 2013; Lovell et al., 2018). The upper Seti Khola gorge below the cirque is a bottleneck prone to blockage by landslides detaching from the cirque walls, and might impound large amounts of water (Kargel et al., 2013). Further downstream, landslides triggered by monsoonal storms (Talchabhadel et al., 2018) could also form temporary dams that might fail catastrophically like in the Melamchi outburst flood in June 2021 (Petley, 2021). Fort (1987) and Kargel et al. (2013) reported that the Sabche Cirque hosts large amounts of unconsolidated material to nourish floods and debris flows. Although not present currently, meltwater lakes could form and grow rapidly in the Sabche Cirque and may release GLOFs in the near future (Zheng et al., 2021). The

Sabche glacier could also contribute to generating outburst floods in the future as its surges could form potentially unstable ice-dammed lakes (Lovell et al., 2018). Despite this evidence of past non-meteorological floods along the Seti Khola, appraisals of flood hazard have so far largely focused on the 100-year meteorological flood as estimated from rainfall data (Basnet and Acharya, 2019; Gurung et al., 2021)."

*General Comment #4:*

The latter seems the most likely to me (also in the light of the 2012 event), but than it is not an 'outburst'. And so I suggest to re-consider and check the use of the word 'outburst' in this context as it could be terminologically misleading (similarly, the use of the word 'risk'). Considering the actual content of the manuscript, my suggestion for possibly revised title would be '(Extreme) flood scenarios and exposed areas in a rapidly growing … ', or similar.

We removed the term "outburst" when referring to the May 2012 flood and changed the title accordingly to the referee's suggestion to "Rare flood scenarios for a rapidly growing high-mountain city: Pokhara, Nepal".

Specific Comments - Text

*Specific Comment #1:*

L99-100: please provide more details on your field mapping of sediment traces; how was it integrated with the overall workflow (Fig. 2)?

Please also refer to our reply to Referee #2's General Comment #7 and their Specific Comment #4.

We mapped sediment traces from orthorectified RapidEye imagery and not in the field; we had stated this in LL162-164 originally: "We also recorded the extent of sediment deposited during the May 2012 flood along this 8.4-km long reach from orthorectified 5-m resolution RapidEye imagery of October 18, 2012." To better clarify our use of sediment traces of the May 2012 flood for model calibration, we now follow this sentence up with "We compared the simulated flood areas with the mapped extent of flood sediments and the mapped changes in buildings." We also added this calibration step to the Fig. 2 and added a similar workflow figure as supplementary material.

*Specific Comment #2:*

L107-108: I wonder what is the justification for using the steady flow HEC-RAS mode while it also offers unsteady flow mode which may be more suitable for this type of events characterised by limited though suddenly released total flood volume and substantial attenuation?

We agree with the referee that unsteady flow modelling might be more suitable to simulate outburst floods. However, we argue that essential data on the flood generating mechanisms in the Sabche Cirque are missing. This lack concerns initial breach characteristics (breach rate, breach depth, impounded water volume, etc.) that are crucial for empirically estimating breach hydrographs. Using

speculative hydrograph as inputs to our models would simply introduce further uncertainties into our inundation maps. To underline this problem, we added the following statement to section 5.1 Inundation Modelling: "While the geomorphic setting caters to numerous potential flood-water sources and generating mechanisms, it lacks any evidence of initial breach characteristics. Yet these unknown parameters are essential for the empirical estimation of hydrographs that are required in models of unsteady flow; hence we restricted our scenario simulations to one-dimensional steady flow."

*Specific Comment #3:*

> L126-127: this is not clear – did you use your field cross-profiles to enhance (manipulate) ALOS DEM? Or how these two are integrated in the study? Please provide more details on your methodology

We changed LL126-129 as follows:

"Geometric data for our HEC-RAS runs were mainly derived from the commercial ALOS 3D digital elevation model (AW3D DEM), which has a vertical and horizontal resolution of <5 m and was projected to UTM Zone 44N (Fig. 2). We also acquired field data with a TruPulse 360 laser range finder and a Garmin eTrex handheld GPS during two campaigns in October of 2016 and 2019. We used these field data to correct some of the 572 DEM-derived channel cross-sections of the Seti Khola and its major tributaries, especially in the narrow gorges."

*Specific Comment #4:*

> L138-146: please consider summarizing these LC classes in table rather than in the text

We thank the referee for this suggestion and accordingly summarised the information on LC and LU classes formerly provided in LL138-146 in a new table.

*Specific Comment #5:*

> L156-158: this is confusing; why don't you name the hazard classes according to the peak discharge?

Please also refer to our replies to Specific Comment Table 1 and Specific Comment Fig. 8.

We underline the difference between total areas covered by a given peak discharge scenario and our definition of hazard classes by expanding this paragraph as follows: "We used a geospatial overlay of our modelled flood inundation boundaries with the LULC data to assess on a semi-quantitative basis the likely impacts of ten peak discharge scenarios. We defined ten flood hazard classes by assigning areas and objects of the smallest flood scenario ($Q_p = 1,000$ m³ s$^{-1}$) to the highest-hazard class (HC) 10. Conversely, the lowest-hazard class 1 is attributed to areas and objects that would be inundated by the largest floods only ($Q_p = 10,000$ m³ s$^{-1}$). In contrast to the total area flooded by a given scenario, these hazard classes do not overlap but label the extent, which a specific scenario covers in contrast to the next lower peak discharge scenario. Hence, HC1 defines the areas and objects that

are potentially impacted by a flood with Qp >= 10,000 m³ s⁻¹, whereas HC10 defines areas and objects that would be inundated by a flood with Qp >= 1,000 m³ s⁻¹."

*Specific Comment #6:*

> L164-165: it would be good to map sediment extent also before the 2012 event, so you could display the change of sediment extent associated with the 2012 flood in your Fig. 3

We thank the referee for this suggestion. We added the sediment extents mapped from RapidEye satellite imagery acquired in March 2012 (22/03/2012) to Figure 3.

*Specific Comment #7:*

> L171-172: simulated peak discharge in Kharapani is 3,700 m3/s, but on L41 you mentioned peak discharge 8,400 m3/s; please comment on this discrepancy

The peak discharge of 3,700 m³ s⁻¹ mentioned in LL171-172 is the result of our model calibration based on sediment traces (also see our replies to Specific Comment #1 and Referee #2's General Comment #7 and their Specific Comment #4) while the peak discharge of 8,400 m³ s⁻¹ cited in LL42 of the introduction was empirically estimated by Oi et al. (2014).

*Specific Comment #8:*

> L282: ok, here you mention that you manipulated the DEM – please provide more details in the methods section

Please refer to our reply to Specific Comment #3.

*Specific Comment #9:*

> L289-290: ok, here is the possible answer to geometric clusters, but would that really be captured by HEC-RAS in this way (can it capture subsurface drainage in this mode)?

Indeed, subsurface drainage is not modelled in our approach as we had stated in LL291-293 originally: "Modelling the groundwater flow in these potential karst structures is beyond the scope of this study and would require hydrological details that remain largely unresolved. We thus modelled flood flows for an idealized gorge geometry informed by DEM and field data." Thus, we describe these locations of flooding in low-lying areas as **artefacts** in the model and explicitly state that: "Hence, interpretation of these model artefacts has to be handled with care." (LL293). As water flow is modelled step-wise from one cross-section to the next, these artefacts potentially occur at cross-sections where the idealised gorge geometry does not capture (or at least underestimates) the available cross-sectional area of flow in the subsurface. This resulting "overspill" of the simulated flow to low-lying areas of the cross-section outside of the idealised gorge produces the model inundation artefacts. Karst features including caves and sinkholes occur in this central part of the

valley, were they are linked to the calcareous deposits of the Ghachok Formation (Fort, 2010; Rimal et al., 2018). We argue that unknown subsurface karst features might cause mismatches in cross-sectional flow area between our idealised model geometry and subsurface conditions. This problem is stated in LL288-291: "Our HEC-RAS simulations also show apparent flooding in places that are likely artefacts of the poorly resolved geometry of several narrow gorges and some subsurface drainage (Fig. 4). Sinkholes and caves in the Ghachok Formation may route some of the discharge of the Seti Khola below the surface, especially in urban areas (Fort, 2010; Rimal et al., 2015; Stolle et al., 2019)". However, we **do not** suggest that the location of the model artefacts in our inundation maps highlights unknown subsurface drainage paths, which still await geotechnical mapping. We further stress this by changing the text passage following LL296 of the original manuscript version: "We thus modelled flood flows for an idealized gorge geometry informed by DEM and field data. Hence, model artefacts potentially occur at cross-sections where this idealised gorge geometry does not capture (or underestimates) subsurface flow; hence, interpretations of the resulting highly localised inundation areas have to be handled with care."

*Specific Comment #10:*

> L337: the recommendations outlined here are very general (probably applicable anywhere) and not really providing a solution for the maintenance failure issue with the previous EW system; moreover, they are in detail elaborated by Thapa et al. (2022); I suggest revision or removal

We altered the text in LL343ff accordingly: "Thapa et al. (2022) pointed out that existing evacuation routes from Yamdi and Kaseri settlements towards higher ground are inadequate and outlined several structural (e.g. river embankment) and non-structural (e.g. evacuation rehearsal drills, population relocation) mitigation measures. The authors proposed to install weather and hydrological stations and to transmit early warnings via mobile-phone, however, their flood response strategy strictly focuses on urban areas in Pokhara's north-west. We argue, that a regularly maintained warning system might want to provide full coverage of all settlements along the Seti Khola's course through the Pokhara valley – including rural settlements and exposed infrastructure in the northern valley. The May 2012 flood also demonstrated that such flash floods can travel fast in the steep headwaters of the Seti Khola (Kargel et al., 2013; Oi et al., 2014). Stream gauges and monitoring stations may need to be located further upstream than presently implemented, ideally close to the outlet of Sabche Cirque to maximise warning times for downstream communities. Complementing Thapa et al.'s (2022) proposal for early-warning strategies in the downstream reaches, we argue that a special focus should be laid on monitoring potentially hazardous developments in the Seti Khola's headwaters. Ideally, continued remote monitoring of the Sabche Cirque using optical and SAR satellite data acquired at short repeat rates afforded via the Sentinel and Planet platforms might further assist early warning while ground movement and deformations of the cirque's walls, surge-phases of the Sabche glacier, and lake formation might be tell-tale warning signs (Grebby et al., 2021; Hermle et al., 2021; Kirschbaum et al., 2019; Quincey et al., 2005). Based on the descriptions by Zhang and Wang (2022) of mitigation efforts of outburst hazard from Lake Cirenmaco, China, monitoring on the ground could include the real-time transmission of optical and thermal data, captured by 360° cameras, to a round-the-clock-operating data processing centre. Yet, the potential cloudiness of this area together with increasing pollution may constrain visual monitoring. Urban planning does not seem a priority for the local government and authorities (Gurung et al., 2021):

instead of relocating people from the vulnerable informal settlements (Yamdi and Kaseri) to safer places, the government and local authorities rather seem to have encouraged people to live in the floodplain by providing them basic amenities (drinking water, electricity, access road) and land owner certificates, thus letting people to live in the riverbanks and flood plains permanently (Gurung et al., 2021)."

Specific Comments – Tables

*Specific Comment Table 1:*

it is a little mystery why there is a fluctuation (not gradual increase) in exposed areas for some of the LULC classes (for instance developed-medium: 153,446 m2 for HC1; 30,214 m2 for HC5; 50,432 m2 for HC6; 18,408 m2 for HC8 and 49,981 for HC10); in other words – why is the impact area of certain LULC not always the highest for the largest peak discharge scenario? Or does it show a difference to the previous peak discharge scenario (hazard class)? Please clarify.

Please also refer to our reply to Specific Comment Fig. 8.

To better clarify the definition of our hazard classes, we added the following explanation to section 3.4: "We used a geospatial overlay of our modelled flood inundation boundaries with the LULC data to assess on a semi-quantitative basis the likely impacts of ten peak discharge scenarios. We defined ten flood hazard classes by assigning areas and objects of the smallest flood scenario ($Q_p$ = 1,000 m³ s$^{-1}$) to the highest-hazard class (HC) 10. Conversely, the lowest-hazard class 1 is attributed to areas and objects that would be inundated by the largest floods only ($Q_p$ = 10,000 m³ s$^{-1}$). In contrast to the total area flooded by a given scenario, these hazard classes do not overlap but label the extent, which a specific scenario covers in contrast to the next lower peak discharge scenario. Hence, HC1 defines the areas and objects that are potentially impacted by a flood with $Q_p$ >= 10,000 m³ s$^{-1}$, whereas HC10 defines areas and objects that would be inundated by a flood with $Q_p$ >= 1,000 m³ s$^{-1}$."

To stress this in Table 1, we also added ">=" to its left column (e.g. HC8, $Q_p$ >= 3,000 m³ s$^{-1}$).

*Specific Comment Table 2:*

please unify the naming of your hazard classes with the rest of the manuscript

We thank the referee for this comment and adjusted the hazard class naming in the first column accordingly.

Specific Comments – Figures

*Specific Comment Fig. 1:*

please consider displaying topography (basic contour lines or cross-profiles at selected locations) in this figure

We thank the referee for this suggestion and opted to add contour lines to our Figure 1.

*Specific Comment Fig. 6:*

please show the complete SE bank in this figure (so the flood extents do not terminate in the air)

We thank the referee for this suggestion and extended the cross-section shown in Figure 6 accordingly.

*Specific Comment Fig. 7:*

the inset map seems to display something different to the main map (or some LULC classes are not shown for some reason (?))

In the original Figure 7's insert only LULC cover categorized in a hazard class is shown while areas with "no hazard" remain white. We understand how this might be confusing. Also following the recommendations of Referee #2 regarding this figure, we changed its design for a better understanding and also match the insert with the main map.

*Specific Comment Fig. 8:*

this is confusing and hard to read and I'm not sure this is the correct graph type to be used; what is the link between this graph and Table 1? What is the actual meaning of the area on y axis (part (a))? For instance, when I sum up the areas for exposed grassland and all scenarios (Table 1), the total area is something about 2,4 km2; here the total area of exposed grassland approx. 17 km2 which is confusing; the largest peak discharge scenario is shown at the bottom (I would expect starting with the 1,000 m3/s, so it can be deduced how much area is exposed for individual scenarios);

Figure 8 shows the **total area** of a given LULC class that is covered by inundation of a **given peak discharge scenario**. Hence, areas covered already in the 1,000 m³/s scenario will also be covered by the scenarios with higher magnitudes. Values for the hazard class areas listed in Table 1 solely depict the area of the zone that this specific scenario covers in contrast to the next smaller peak discharge scenario (area of **difference** between the scenarios' inundation limits). In contrast to total inundated areas covered by a certain peak discharge scenario, HC zones do not overlap. For example, HC 4 defines the zone, which will potentially not be affected by floods with peak discharges smaller than 7,000 m³ s⁻¹ but by floods with a peak discharge of 7,000 m³ s⁻¹ and larger.

We now better underline this difference in definition between total areas covered in a given peak discharge scenario and the hazard class zones in 3.4: "We used a geospatial overlay of our modelled flood inundation boundaries with the LULC data to assess on a semi-quantitative basis the likely impacts of ten peak discharge scenarios. We defined ten flood hazard classes by assigning areas and objects of the smallest flood scenario (Qp = 1,000 m³ s⁻¹) to the highest-hazard class (HC) 10. Conversely, the lowest-hazard class 1 is attributed to areas and objects that would be inundated by the largest floods only (Qp = 10,000 m³ s⁻¹). In contrast to the total area flooded by a given scenario, these hazard classes do not overlap but label the extent, which a specific scenario covers in contrast

to the next lower peak discharge scenario. Hence, HC1 defines the areas and objects that are potentially impacted by a flood with Qp >= 10,000 m³ s⁻¹, whereas HC10 defines areas and objects that would be inundated by a flood with Qp >= 1,000 m³ s⁻¹."

*Specific Comment Fig. 10 and 11:*

Figs. 10 and 11: what are those strange geometric linear clusters of buildings exposed to hazard class 1-3? What about that cluster located far (and disconnected) from the river?

These clusters are potential model artefacts and, hence, we refer the referee to our reply to Specific Comment #9.

**References**

[revised manuscript text omitted]

---

## Author Comment (AC2)

Reply to Referee Comment #2 by an Anonymous Referee

General Comments

*General Comment #1:*

As I can see, a fair bit of work has been carried out, however, the current form of the manuscript lacks sufficient novelty and several vital information. Moreover, the usage of a few terminologies such as 'outburst', and 'risk' need justification. Details on the numerical aspects of the flood model set-up, which is vital in justifying the impact assessment are also missing in the text.

We thank the referee for their evaluation and refer them to our replies to General Comments #5, #8, and #10.

*General Comment #2:*

1. Introduction- In the current form, the introduction projects more or less about the study region, and flood incidences. It is understandable that the focus of the study is on a mountainous region, however, following a generic (or top-down) approach to flood risk, and other flood-related issues may be desirable. A few statistics on concomitant climate change impacts may also be added here to show the severity of the flooding events.

We agree with the referee that a more generic introduction to flood risk in a globally changing climate might be interesting and are thankful for this suggestion. However, as we submitted our manuscript to the special issue "Estimating and predicting natural hazards and vulnerabilities in the Himalayan region" we prefer to keep our introduction focussed on this region in general and the Pokhara valley as our study area in particular. We added information on projected climate change impacts on the Hindu-Kush Himalayas starting from LL31: "Current projections of cryospheric change in this mountain belt include a continued total glacier-mass wastage of up to 64 ± 5% by the end of the century under the RCP8.5 scenario (Kraaijenbrink et al., 2017) as well as permafrost degradation (Bolch et al., 2019). These changes will likely result in a destabilisation of mountain slopes and increase in meltwater volumes stored in lakes impounded behind potentially instable natural dams (Hock et al., 2019). Thus, the potential for hazards caused from these instabilities, including sudden floods, is likely to increase in the future. The Pokhara valley in Nepal, home to the nation's second largest city, is a prime example of such a Himalayan valley with rapid socio-economic development: […]."

*General Comment #3:*

3. Section 3.2- Not enough justification is provided on the selection of the ten peak discharge scenarios as inputs to the HEC-RAS model. Moreover, why did the authors consider a range between 1,000 and 10,000 m3/s? Please elaborate.

We ask the referee to refer to our reply to Referee #1's General Comment #2 regarding our justification of the tested range of peak discharge scenarios.

*General Comment #4:*

> 4. In continuation to the previous query, a major discrepancy arises with the class intervals (1000 m3/sec) between each peak discharge. What if there is a peak-discharge falling in the mid-way of two end values, which may not be incorporated appropriately within the flood model, but will add up the impacts on the communities.

We are unsure what the referee means by "major discrepancy". The scenarios that we present here collapse a number of assumptions into a few scenarios. By definition, none of these (or any other scenario) can account for all possible outcomes. Instead, the scenarios offer some general and consistently structured insight into the extent of flooding given for a range of peak discharges. We are interested here in a first-order appraisal of how inundated area scales with peak discharge instead of providing detailed flood-prediction maps for every possible flood size.

*General Comment #5:*

> 8. Details of the time step of the HEC-RAS model simulation, final resolution of flood inundation maps, and courant number must also be added in section 3.2. Further, the justification of considering ALOS DEM (which is a freely available global product) as the bathymetry map for the study area may also be added, as sensitivity (if any) from the DEM will be reflected as inaccuracies in the set of flood inundation maps.

We added information on the resolution of inundation maps and the used ALOS DEM.

The 5-m ALOS 3D enhanced DEM is a commercial product and has the highest available resolution for our study area. Yet especially the narrow gorges of the study area are not well resolved such that we manually corrected cross-sections with field data (see our reply to Referee #1's Specific Comment #3). We agree with the referee's statement that DEM resolution is a constraint on the accuracies of the flood inundation maps, but had explicitly acknowledged this in the original manuscript version: "The accuracy of our results hinges on the accuracy of river cross-sections and the estimates of channel and overbank roughness (Manning's *n*; Westoby et al., 2014; Wohl, 1998). Previous studies of HEC-RAS for outburst floods have used mostly coarser digital elevation data than the 5-m ALOS DEM we used here (Mergili et al., 2011; Somos-Valenzuela et al., 2014; Wang et al., 2018; Zhang and Liu, 2015). The stereo satellite imagery forming the basis for this DEM was acquired between 2006 and 2011 and excludes channel changes by the May 2012 flood (Gurung et al., 2021). We minimised potential resulting effects on our models by manually adjusting cross-sections with our additional field-surveyed elevation data." (LL277-282).

Further, we model one-dimensional steady state flow and, thus do not specify time steps or a Courant number in our HEC-RAS models. The necessary model inputs for this approach (Manning's *n*, upper and lower boundary conditions, baseflow in the tributaries) were accounted for in our original manuscript in LL129-136. Please also refer to our reply to Referee #1's Specific Comment #2 for our reason for avoiding unsteady-state modelling.

*General Comment #6:*

> 10. The description of "Hazard" in the manuscript is ambiguous. Hazard indicates the severity of an event and is usually represented in terms of floodwater depth, velocity, the residence time of floodwater, etc. As a result of which, directly attaching the discharge scenarios to different levels of hazards is a very preliminary attempt. In another way, authors might consider terming them as low to high hazard classes rather than providing hazard classes as such.

Although our models yield information on floodwater depth and velocity (LL181, 186, 188 and Fig. 5, Fig. 6), we refrained from using these as metrics of inundation hazard due to the large number of tested scenarios. Classifying flood hazard based on, for example, inundation depths for ten different peak discharge scenarios each is arguably more detailed, but also much harder to communicate. The main aim of our paper was to raise awareness that large, rarer non-meteorological floods, which are so far hardly considered in hazard assessments, may affect several rapidly expanding settlements and infrastructure. Thus, we deliberately opted for a simple approach and classified hazard based on whether a given area or object would lie within the inundation limits of a given flood magnitude on the basis that higher flood peak discharges are exceeded more infrequently. Hence, objects and areas within the inundation limits of smaller peak discharge scenarios are more likely to be affected and are, thus, attributed to a higher-hazard class.

*General Comment #7:*

> 12. How did the authors carry out calibration and validation of the flood inundation outputs? Without this, the impact assessment over various land-use classes does not seem fitting.

Please also refer to our reply to Specific Comment #4 and Referee #1's Specific Comment #1.

We had stated in the original manuscript that "Stream-gauge data are unavailable for the Seti Khola, hence we validated our model with mapped damage and sediment traces caused by the May 2012 flood." (LL100-101). In the absence of measurements, we use the extents of fluvial sediment deposition during the May 2012 flood as palaeo-stage indicators of the maximum inundation limits. We applied step-backwater hydrodynamic modelling to the 2012 flood and proceeded with this calibrated model to simulate inundation limits of our ten peak discharge scenarios.

*General Comment #8:*

> 13. At several places in the manuscript, the term 'outburst' flood appears misleading as there is no mention of the temporal dynamics of the flooding event. I request the authors to either justify or remove the 'outburst' term wherever it is mentioned in the manuscript.

We agree with the referee that our simulations do not include a temporal component due to the modelling restrictions described in detail in response to Referee #1's Specific Comment #2. We use the term "outburst flood" to refer to the geomorphic process chain, in which a quantity of water,

which was retained behind a natural dam, is released and propagates downstream (Costa and Schuster, 1988). The term "outburst flood", thus, distinguishes this process (chain) from meteorologically-triggered flash floods. We had stressed in LL325-335, and now also in the introduction (see our reply to Referee #1's General Comment #1 and #3), that geomorphic and glaciologic activity in the Sabche Cirque could promote the rapid forming and growth of water bodies (Kargel et al., 2013). Our inundation maps quantify the downstream inundation, i.e. the eventual impacts of this specific process. We now have more comprehensively underlined our study focus on these non-meteorological floods in the introduction, and removed the term "outburst" to be less exclusive about the flood-generating processes involved.

*General Comment #9:*

14. The list of recommendations provided in the manuscript is very generic and applicable to any other case study. I suggest the authors be very specific and structure this section into possible structural and non-structural recommendations for flood management.

We wish to refer the referee to our reply to Referee #1's Specific Comment #10 regarding the recommendations section.

*General Comment #10:*

15. I am not fully convinced with the title of the manuscript over two points- 'outburst', and 'risks'. The query regarding the usage of the former terminology is already mentioned in one of my earlier comments. The manuscript actually does not quantify 'risk', as it does not account for vulnerability as such. The impact assessment addressed in the work is more of an exposure assessment. Therefore, the usage of 'risk' terminology may be avoided in the title and elsewhere in the text. Moreover, the hazard is quantified as the extent of the inundated area, which is a very simple form of indicating a flood hazard.

Please refer to our reply to General Comment #8 regarding the use of the term "outburst flood". Following Referee #1's suggestion, we change the title to "Rare flood scenarios for a rapidly growing high-mountain city: Pokhara, Nepal". We also removed the term "risk" when referring to our assessment and replace it with "exposure".

Specific Comments - Text

*Specific Comment #1:*

Line 124: Vertical resolution of ALOS-DEM should be mentioned.

We altered LL124-125 as follows: "Geometric data for our HEC-RAS runs were mainly derived from the commercial ALOS 3D digital elevation model (AW3D DEM), which has a vertical and horizontal resolution of <5 m and was projected to UTM Zone 44N (Fig. 2)."

*Specific Comment #2:*

Line 127: The authors mention the consideration of around 572 cross-sections of the river channel. A separate figure providing these details may be provided, if possible in the supplementary material.

We thank the referee for this suggestion and added a map displaying the river geometry, i.e. cross-sections, main and tributary reaches, and junctions, used for our modelling in HEC-RAS as supplementary material.

*Specific Comment #3:*

Line 141: The description of the land-use classes is not required to be added to the text. This may be provided in the form of a separate figure in the supplementary material.

Following Referee #1's suggestion, we summarised the information on LC and LU classes formerly provided in LL139-146 in a new table.

*Specific Comment #4:*

Line 164: How was the extent of sediment deposition quantified for the May 2012 flood event from the satellite imagery? How this piece of information was useful to the research addressed in this manuscript? Please justify.

Please also refer to our reply to General Comment #7 and Referee #1's Specific Comment #1.

The May 2012 flood represents the only documented flood event to validate our results on. However, as stated in LL100-101, stream-gauge data for the May 2012 flood are unavailable and we had to validate our model with sediment and damage traces, which we interpret as proxys of the inundated area. We manually delineated the extents of flood deposits from orthorectified 5-m resolution RapidEye images acquired on October 18, 2012 - the first cloud-free image following the flood. The sediments deposited by the flood stood out as bright pixels on the otherwise densely vegetated terraces. This use of sedimentary evidence or paleo-stage indicators to reconstruct peak discharges of floods occurring in ungauged river streams has been applied by a number of authors. Retrospective step-backwater hydrodynamic modelling of one-directional steady flow in HEC-RAS – as presented in our study - has been, for example, successfully applied in the Himalayas by Cenderelli and Wohl (2001) or in the Andes by Klimeš et al. (2014). Please also refer to our reply to General Comment #3.

Specific Comments – Figures

*Specific Comment Fig. 1:*

Please add an appropriate legend to describe what the triangles (stations) represent. An inset map of the elevation\topography of the study area may be included within this figure as well.

We thank the referee for this suggestion and added a legend accordingly (grey triangles mark the locations of settlements mentioned in this study). Following the suggestions of Referee #1, we added contour lines to visualise the topography of our study area.

*Specific Comment Fig. 7:*

The description of hazard classes within various land use classes is very difficult to locate. Some sort of different representation may be thought of here to locate the degree of hazard distinctly within land-use classes or create a separate figure for the same.

We adjusted the figure accordingly to improve its clarity.

References

Bolch, T., Shea, J. M., Liu, S., Azam, F. M., Gao, Y., Gruber, S., Immerzeel, W. W., Kulkarni, A., Li, H., Tahir, A. A., Zhang, G. and Zhang, Y.: Status and Change of the Cryosphere in the Extended Hindu Kush Himalaya Region BT - The Hindu Kush Himalaya Assessment: Mountains, Climate Change, Sustainability and People, edited by P. Wester, A. Mishra, A. Mukherji, and A. B. Shrestha, pp. 209–255, Springer International Publishing, Cham., 2019.

Cenderelli, D. A. and Wohl, E. E.: Peak discharge estimates of glacial-lake outburst floods and "normal" climatic floods in the Mount Everest region, Nepal, Geomorphology, 40(1–2), 57–90, doi:10.1016/S0169-555X(01)00037-X, 2001.

Costa, J. E. and Schuster, R. L.: The Formation and Failure of Natural Dams, Bull. Geol. Soc. Am., 100(7), 1054–1068, doi:10.1130/0016-7606(1988)100<1054:TFAFON>2.3.CO;2, 1988.

Gurung, N., Fort, M., Bell, R., Arnaud-Fassetta, G. and Maharjan, N. R.: Hydro-torrential hazard vs. anthropogenic activities along the Seti valley, Kaski, Nepal: Assessment and recommendations from a risk perspective, J. Nepal Geol. Soc., 62, 58–87, doi:10.3126/jngs.v62i0.38695, 2021.

Hock, R., Rasul, G., Adler, C., Cáceres, B., Gruber, S., Hirabayashi, Y., Jackson, M., Kääb, A., Kang, S., Kutuzov, S., Milner, A., Molau, U., Morin, S., Orlove, B. and Steltzer, H. I.: High Mountain Areas, in IPCC Special Report on the Ocean and Cryosphere in a Changing Climate, edited by H.-O. Pörtner, D. C. Roberts, V. Masson-Delmotte, P. Zhai, M. Tignor, E. Poloczanska, K. Mintenbeck, A. Alegría, M. Nicolai, A. Okem, J. Petzold, B. Rama, and N. M. Weyer, pp. 131–202, Genf., 2019.

Kargel, J. S., Paudel, L., Leonard, G., Regmi, D., Joshi, S., Poudel, K., Thapa, B., Watanabe, T. and Fort, M.: Causes and human impacts of the Seti River (Nepal) disaster of 2012, in Glacial Flooding & Disaster Risk Management Knowledge Exchange and Field Training; High Mountains Adaptation Partnership: Huaraz, Peru, pp. 1–11, Huaraz., 2013.

Klimeš, J., Benešová, M., Vilímek, V., Bouška, P. and Cochachin Rapre, A.: The reconstruction of a glacial lake outburst flood using HEC-RAS and its significance for future hazard assessments: An example from Lake 513 in the Cordillera Blanca, Peru, Nat. Hazards, 71(3), 1617–1638, doi:10.1007/s11069-013-0968-4, 2014.

Kraaijenbrink, P. D. A., Bierkens, M. F. P., Lutz, A. F. and Immerzeel, W. W.: Impact of a global temperature rise of 1.5 degrees Celsius on Asia's glaciers, Nature, 549(7671), 257–260, doi:10.1038/nature23878, 2017.

Mergili, M., Schneider, D., Worni, R. and Schneider, J. F.: Glacial lake outburst floods in the Pamir of Tajikistan: Challenges in prediction and modelling, Int. Conf. Debris-Flow Hazards Mitig. Mech. Predict. Assessment, Proc., 973–982, doi:10.4408/IJEGE.2011-03.B-106, 2011.

Somos-Valenzuela, M. A., McKinney, D. C., Byers, A. C., Rounce, D. R., Portocarrero, C. and Lamsal, D.: Assessing downstream flood impacts due to a potential GLOF from Imja Lake in Nepal, Hydrol. Earth Syst. Sci. Discuss., 11(11), 13019–13053, doi:10.5194/hessd-11-13019-2014, 2014.

Wang, W., Gao, Y., Iribarren Anacona, P., Lei, Y., Xiang, Y., Zhang, G., Li, S. and Lu, A.: Integrated hazard assessment of Cirenmaco glacial lake in Zhangzangbo valley, Central Himalayas, Geomorphology, 306, 292–305, doi:10.1016/j.geomorph.2015.08.013, 2018.

Westoby, M. J., Glasser, N. F., Brasington, J., Hambrey, M. J., Quincey, D. J. and Reynolds, J. M.: Modelling outburst floods from moraine-dammed glacial lakes, Earth-Science Rev., 134, 137–159, doi:10.1016/j.earscirev.2014.03.009, 2014.

Wohl, E. E.: Uncertainty in Flood Estimates Associated with Roughness Coefficient, J. Hydraul. Eng.,

124(2), 219–223, doi:10.1061/(asce)0733-9429(1998)124:2(219), 1998.

Zhang, X. and Liu, S.: A framework of numerical simulation on moraine-dammed glacial lake outburst floods, J. Arid Land, 7(6), 728–740, doi:10.1007/s40333-015-0133-x, 2015.